# Microbiota derived short chain fatty acids promote histone crotonylation in the colon through histone deacetylases

Rachel Fellows[1], Jérémy Denizot[1,9], Claudia Stellato[1], Alessandro Cuomo[2], Payal Jain[1], Elena Stoyanova[1], Szabina Balázsi[1], Zoltán Hajnády[1], Anke Liebert[1], Juri Kazakevych[1], Hector Blackburn[1], Renan Oliveira Corrêa[3], José Luís Fachi[3], Fabio Takeo Sato[3], Willian R. Ribeiro[4,5], Caroline Marcantonio Ferreira[4], Hélène Perée[1], Mariangela Spagnuolo[1], Raphaël Mattiuz[1], Csaba Matolcsi[1], Joana Guedes[6], Jonathan Clark[7], Marc Veldhoen[6,10], Tiziana Bonaldi[2], Marco Aurélio Ramirez Vinolo[3] & Patrick Varga-Weisz[1,8]

The recently discovered histone post-translational modification crotonylation connects cellular metabolism to gene regulation. Its regulation and tissue-specific functions are poorly understood. We characterize histone crotonylation in intestinal epithelia and find that histone H3 crotonylation at lysine 18 is a surprisingly abundant modification in the small intestine crypt and colon, and is linked to gene regulation. We show that this modification is highly dynamic and regulated during the cell cycle. We identify class I histone deacetylases, HDAC1, HDAC2, and HDAC3, as major executors of histone decrotonylation. We show that known HDAC inhibitors, including the gut microbiota-derived butyrate, affect histone decrotonylation. Consistent with this, we find that depletion of the gut microbiota leads to a global change in histone crotonylation in the colon. Our results suggest that histone crotonylation connects chromatin to the gut microbiota, at least in part, via short-chain fatty acids and HDACs.

[1] Nuclear Dynamics, Babraham Institute, Cambridge, CB22 3AT, UK. [2] Department of Experimental Oncology, Istituto Europeo di Oncologia, 20139 Milano, Italy. [3] Laboratory of Immunoinflammation, Institute of Biology, UNICAMP, Campinas 13083-862, Brazil. [4] Department of Pharmaceutical Sciences, Institute of Environmental, Chemistry and Pharmaceutical Sciences, Universidade Federal de São Paulo, Diadema, SP 09913-03, Brazil. [5] Chemical Biology Graduate Program, Universidade Federal de São Paulo, Diadema SP 09913-03, Brazil. [6] Lymphocyte Signalling and Development, Babraham Institute, Cambridge, CB22 3AT, UK. [7] Biological Chemistry, Babraham Institute, Cambridge CB22 3AT, UK. [8] School of Biological Sciences, University of Essex, Colchester CO4 3SQ, UK. [9] Present address: Université Clermont Auvergne, Inserm U1071, INRA USC2018, M2iSH, Clermont–Ferrand F-63000, France. [10] Present address: Instituto de Medicina Molecular, Faculdade de Medicina da Universidade de Lisboa, Lisbon 1649-028, Portugal. Jérémy Denizot, Claudia Stellato, Alessandro Cuomo, Payal Jain and Elena Stoyanova contributed equally to this work. Correspondence and requests for materials should be addressed to T.B. (email: tiziana.bonaldi@ieo.it) or to M.R.A.V. (email: mvinolo@unicamp.br) or to P.V.-W. (email: patrick.varga-weisz@babraham.ac.uk)

Histone post-translational modifications (HPTMs) are fundamental regulators of gene expression and are tightly controlled by enzymes that respond to the availability of metabolic precursors[1]. Histone acetylation is a well-studied HPTM usually linked to active genes and is added to various lysine groups of histones by histone acetyltransferases (HATs) and removed by histone deacetylases (HDACs). More recently, various longer chain acylations of histones have been characterized, including crotonylation[2], butyrylation[3, 4], and hydroxybutyrylation[5]. These acylations have been linked to cellular metabolism, because they reflect the availability of the short-chain fatty acids (SCFAs) and their coenzyme A adducts in the cell[5, 6] (reviewed in refs. [7, 8]). This has been demonstrated by introducing crotonate (2-butenoate), an SCFA moiety produced intracellularly as an intermediate of metabolic processes[2, 6, 9, 10], to the cell culture media which affects histone crotonylation levels. Histone crotonylation reprograms the functionality of nucleosomes, setting it apart from histone acetylation, by favoring interactions with a specific set of chromatin modifiers[9–12].

A link between cellular metabolism, SCFAs, and transcriptional regulation is particularly relevant in the intestine where microorganisms break down complex carbohydrates to SCFAs such as acetate, propionate, and butyrate[13, 14]. SCFAs are an important component of normal gut physiology by providing a major energy source for the colon epithelial cells[15]. They also affect cellular functions and modulate immune responses, in part by affecting gene expression and the epigenome through inhibiting HDACs[14, 16]. Here, we explore histone crotonylation in intestinal

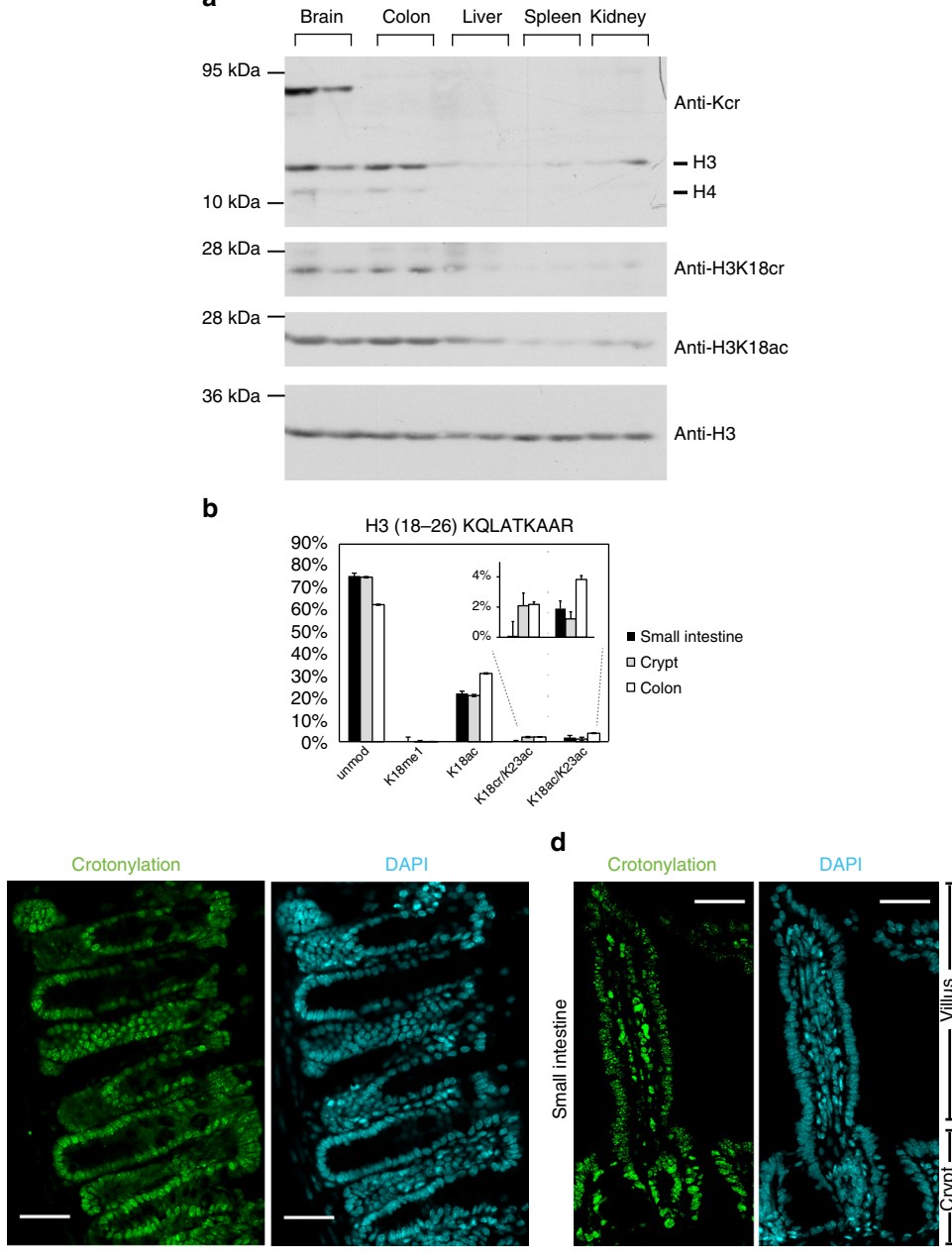

**Fig. 1** Histone crotonylation is found in the intestine. **a** Western blot analysis of whole cell extracts from several mouse tissues using indicated antibodies shows that histone crotonylation is particularly abundant in the brain and colon; the analysis of tissues from two mice is shown. **b** Relative abundance of H3K18cr in the intestinal epithelium cell fractions, $n = 3$, error bars are standard deviation. **c**, **d** Immunofluorescence microscopy with anti-pan crotonyl antibody (green, left panels) and DAPI counterstaining (cyan, right panels) of a mouse colon (**c**) and small intestinal (**d**) tissue sections, scale bars 40 μm

**Table 1 List of K-crotonyl histone-modified peptides**

| Modified sequences | Modified sites | Gene names | Charge | m/z | RT | Score |
|---|---|---|---|---|---|---|
| K(cr)STGGK(*)APR | K9 | HIST2H3 | 2+ | 507.7921 | 71.3 | 60.3 |
| K(*)STGGK(cr)APR | K14 | HIST2H3 | 3+ | 338.8639 | 72.9 | 33.4 |
| K(cr)QLATKAAR | K18 | HIST2H3 | 2+ | 550.3367 | 90.8 | 93.1 |
| K(cr)QLATK(ac)AAR | K18 | HIST2H3 | 2+ | 548.8273 | 91.0 | 89.1 |
| K(cr)SAPATGGVKKPHR | K27 | HIST2H3 | 3+ | 531.3110 | 85.4 | 39.7 |
| YQK(cr)STELLIR | K56 | HIST2H3 | 2+ | 659.8710 | 129.0 | 47.3 |

Summary of identified crotonylation sites on histone H3 from the crypt fraction of the small intestine using MS/MS analysis
SI small intestine, cr crotonylation, ac acetylation, (*) chemical alkylation

epithelial cells and show that histone H3 lysine 18 crotonylation (H3K18cr) is readily detectable in this tissue and that histone crotonylation is regulated by class I HDACs. Our findings suggest that histone crotonylation connects chromatin structure to the gut microbiota via HDACs and SCFAs.

## Results

**Histone crotonylation abundance in the intestine**. Western blot analysis of the level of histone crotonylation in several tissues (colon, brain, liver, spleen, kidney) using the antibodies against crotonyl-lysine and H3K18cr indicates that the greatest levels of histone crotonylation are in colon and, interestingly, brain among the tissues analyzed (Fig. 1a). An approximately 70 kDa protein in the brain extract is recognized by the antibody against crotonyl-lysine, indicating the presence of a crotonylated non-histone protein in the brain.

To characterize the pattern of histone H3 PTMs in intestinal cells, we analyzed small intestine epithelium, colon epithelium, and small intestine crypt-enriched fractions by LC-MS/MS (Supplementary Fig. 1, LC-MS: liquid chromatography-mass spectrometry). This analysis allowed the site-specific identification of 38 differentially modified peptides (Supplementary Table 1). Among them, we identified lysine methylation and acetylation combinations at several H3 peptides and increased levels of monomethylated H3K4 in both small intestinal crypt and colon fraction compared to the fraction from whole small intestine epithelium. Di- and tri-methylated H3K4 were below the detection limit, likely due to their low abundance (Supplementary Fig. 1 and Supplementary Table 1). Most interestingly, we identified histone lysine crotonylation at several histone H3 residues: K9, K14, K18, K27, and K56 (Table 1, Supplementary Fig. 2, and Supplementary Table 1). H3K18 crotonylation (H3K18cr), in association with H3K23 acetylation, was the most abundant histone H3 crotonylation mark in crypt and colon fractions (Fig. 1b, Supplementary Table 1), possibly suggesting that these two marks are co-regulated. H3 crotonylation abundance is overall rather low at K9, K27, and K56. Representative MS/MS spectra for all crotonylated peptides are displayed in Supplementary Fig. 2.

As immunostaining with anti-H3K18cr antibody did not work in our hands, we performed immunostaining of murine small intestine and colon using antibodies targeting crotonyl-lysine (anti-Kcr) and histone H4 crotonylated at K8 (anti-H4K8cr). This demonstrated the presence of these modifications in the nuclei of intestinal epithelium cells, especially in the proliferative crypt compartments (Fig. 1c, d, Supplementary Fig. 3 and 4). Western blot analysis of in vitro crotonylated or acetylated histones and of whole colon extracts confirmed specificity of the anti-Kcr, anti-H3K18cr, and anti-H3K18ac antibodies (Supplementary Fig. 5a, b).

**Genome-wide localization of H3K18cr in the colon epithelium**. As we found that histone H3K18cr is the most abundant histone

crotonylation mark in the intestine, we characterized it further by chromatin immunoprecipitation-sequencing (ChIP-seq). This analysis showed that H3K18cr is associated with transcription start sites (TSS) (Figs. 2a–d), similar to H3K4me3 (Fig. 2c), as has been shown before in macrophages[6]. To investigate the link between H3K18cr and transcription, we performed RNA-sequencing (RNA-seq) on colon epithelial crypts and found higher gene expression levels associated with increased H3K18cr enrichment over TSS (Fig. 2e). KEGG pathway analysis of genes with high levels of H3K18cr over their TSS highlights various pathways, in particular several involved in cancer, suggesting that deregulation of histone crotonylation may be linked to cancer (Fig. 2f, Supplementary Fig. 6).

**Microbiota and histone crotonylation in the mouse colon**. As histone crotonylation has been linked to cellular metabolism and we found it to be relatively abundant in the gut, we hypothesized that this modification may be linked to the SCFAs that are generated by intestinal microbiota, especially in the colon. Treatment of mice for 3 days with a cocktail of antibiotics led to a reduction of the bacterial load (Suppl. Figure 7a) and a reduction of SCFAs in colon luminal content and serum (Fig. 3a). This was linked to a noticeable global decrease of histone crotonylation in the colon tissue, which was particularly clear with histone H4K8 crotonylation but also for H3K18 and H4 crotonylation (as detected by the anti-Kcr antibody) (Figs. 3b, c). Interestingly, we found that the treatment with antibiotics was also linked to increased amounts of HDAC2 (Figs. 3d, c; changes in HDAC1 and HDAC3 levels were not consistent, Supplementary Fig. 7b).

**SCFAs promote histone crotonylation**. Next we investigated if the drop in SCFA concentration in the colon lumen and serum could account for the observed global reduction in histone crotonylation. When added to the media of human colon carcinoma cells (HCT116) and mouse small intestinal organoids, the SCFA crotonate promoted H3 and H4 crotonylation, thereby confirming previous findings[6] (Fig. 4a, Supplementary Fig. 8a). We found that histone crotonylation is highly dynamic, being increased within 2 h of crotonate addition to the medium and lost within 1 h of wash out (Supplementary Fig. 8b). The chemically related SCFA butyrate, which is naturally present in the intestine, also promoted histone crotonylation at physiologically relevant concentrations, both in gut organoids and HCT116 cells (Fig. 4a, Supplementary Fig. 8c). Therefore, these in vitro observations are consistent with the idea that depletion of the microbiota leads to a decrease in histone crotonylation of the colon epithelium because of the drop in SCFAs.

**HDAC inhibitors upregulate histone crotonylation**. Since butyrate is a well-known HDAC inhibitor[17], we hypothesized that HDACs might regulate histone crotonylation and their inhibition explains the increase in histone crotonylation upon butyrate

treatment. We found that other HDAC inhibitors are also able to promote histone crotonylation in HCT116 cells, including trichostatin A (TSA), the clinically relevant suberanilohydroxamic acid (SAHA), and valproic acid (VPA, Supplementary Fig. 9). MS275 (entinostat), a selective inhibitor of class I HDACs, which has been tested in various clinical trials[18], significantly promoted histone crotonylation at sub-micromolar concentrations in

HCT116 cells (Fig. 4b). This outcome cannot be explained by cross-reactivity of the anti-crotonyl antibodies to acetylated histones[6] (Supplementary Fig. 5). A ChIP-qPCR for H3K18cr observed a significant increase in enrichment upon MS275 treatment in different genomic loci including repetitive regions, suggesting that class I HDAC inhibition leads to a spread of histone crotonylation into intergenic areas (Fig. 4c).

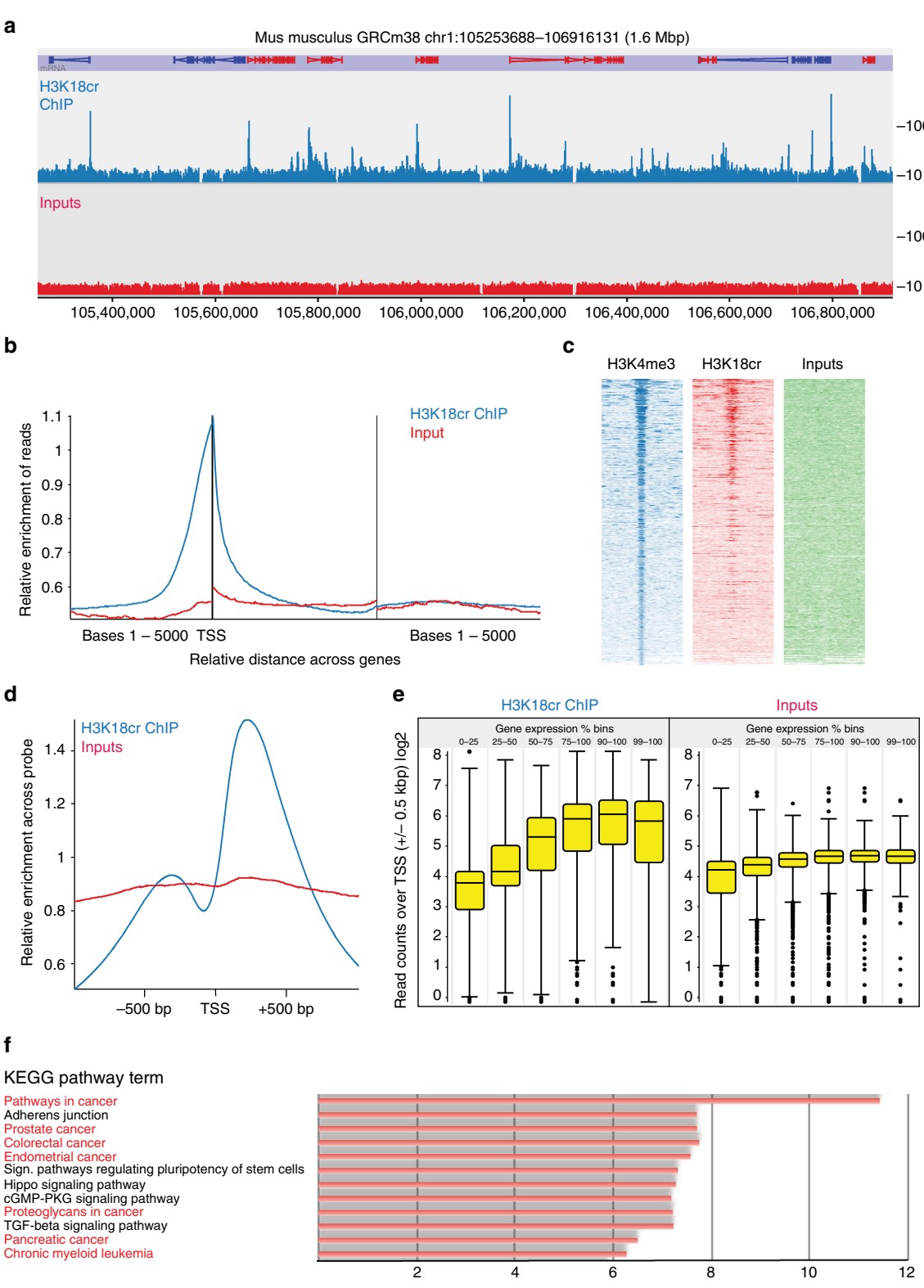

To investigate how HDAC inhibitors change the distribution of histone crotonylation over the genome, we performed ChIP-seq analysis of H3K18cr and H3K18ac. This showed a relative decrease of read counts over TSS when HCT116 cells were treated with MS275 (Fig. 5a, b). This could be explained with an overall increase of these marks over the genome outside of the TSS regions. ChIP-seq measures the relative proportional distribution of a mark over the genome. If the proportional increase of H3K18cr over TSS is lower than the one outside of TSS, an increase of histone crotonylation over regions outside of TSS (which cover a much greater proportion of the genome than all TSS combined) will lead to an apparent proportional drop of this mark in TSS regions.

Interestingly, H3K18cr and H3K18ac peaks correlate highly and occur at the same sites genome-wide in HCT116 cells, with or without MS275 treatment (Figs. 5c–e). MS275 treatment of HCT116 cells caused changes in gene expression, which are linked to changes in H3K18 acetylation and crotonylation (Fig. 5f, g). Thus, gene silencing and chromosome segregation defects occurring upon HDAC inhibitor treatments[19] may not only be due to aberrant histone acetylation, but could also be linked to histone crotonylation.

**Histone crotonylation is linked to the cell cycle**. We performed cell cycle arrest and release experiments in presence or absence of the class I HDAC specific inhibitor MS275. The results indicated that histone crotonylation, including H3K18cr, is linked to cell cycle progression, showing an increase in S and G2-M phase over G1 arrested cells (Fig. 6, lanes 1–8). This experiment also suggests that class I HDACs may be involved in this cell cycle-mediated modulation of histone crotonylation levels, as the G1-arrest-linked downregulation of histone crotonylation is inhibited in the presence of MS275 compared to untreated cells (Fig. 6, lanes 1, 2, 9, 10). In addition, in MS275-treated cells, histone crotonylation was modulated during the cell cycle, but levels remained generally higher compared to untreated cells (Fig. 6, lanes 9–16).

**Class I HDACs are histone decrotonylases**. To test further if we could link class I HDACs to decrotonylation, we transiently over-expressed a fusion protein of HDAC1 with green fluorescent protein (GFP), an N-terminal deletion mutant version of this or GFP in HCT116 cells, sorted the transfected cells, and monitored histone crotonylation by western blot analysis. This showed that the over-expression of HDAC1 but not its mutant version with an N-terminal truncation (or GFP alone) caused some reduction of global histone H3K18 and H4K8 crotonylation (Supplementary Fig. 10).

To validate the ability of HDACs to decrotonylate histones, we tested this activity with recombinant, purified HDACs in in vitro experiments and found that the class I HDAC enzymes HDAC1, HDAC2, and the HDAC3/Ncor1 complex efficiently removed the crotonyl moiety of in vitro crotonylated histones (Fig. 7a). Using

this in vitro approach, we estimated enzyme kinetic values for the decrotonylation of H3K18cr and deacetylation of H3K18ac and found that HDAC1 has a similar capacity to perform each reaction. HDAC1 has lower $V_{max}$ and $K_{cat}$ values for the decrotonylation compared to the deacetylation reactions indicating that HDAC1 has a higher maximum turnover with the acetyl substrate than the crotonyl substrate (Fig. 7b, Supplementary Fig. 11). We found that the $K_m$ with acetylated histones of 1.47 µM was similar to a published $K_m$ of HDAC with a BOC-lys (acetyl)-AMC of 3.7 µM[20]. Published $K_m$ values with different fluorescent or tritiated acetyl substrates vary from 0.68 to 78 µM[21–24]. Relative to this, the $K_m$ of the crotonyl substrate was lower at 0.42 µM suggesting that HDAC1 can efficiently bind crotonylated histones at low concentrations of substrate. This may be relevant in the context of the cell where HDAC1 could respond to minor fluctuations in the availability of the crotonyl substrate.

In vitro, HDAC inhibitors TSA and butyrate reduced decrotonylation by HDAC1 significantly, demonstrating that the HDAC inhibitors, including SCFAs, affect decrotonylation by HDACs directly (Fig. 7c). TSA inhibited deacetylation at lower doses (IC50 ~5 nM) than decrotonylation (IC50 ~50 nM) highlighting the differential impact of inhibitors on HDAC activity and potential for specific drug development. We also found that crotonate inhibited decrotonylation and deacetylation in vitro (Fig. 7c). Remarkably, at crotonate concentrations of 6 mM or higher, HDAC1 catalyzed the addition of the crotonyl moiety to lysine groups of histones (Fig. 7d). This did not occur with acetate or butyrate, possibly due to their lower chemical reactivity compared to crotonate. While this finding may not be directly biologically relevant, as it is unlikely that such high concentrations of crotonate are found intracellularly, it highlights the reversible nature of the decrotonylation reaction by HDAC1.

A previous study dismissed HDACs 1–11 as decrotonylases based on an in vitro fluorometric assay using a peptide analog[2]. We also did not find decrotonylase activity for HDAC1 using the same assay, but found that the BOC-Lys(crotonyl)-AMC compound used is inhibitory for the deacetylation step of the BOC-Lys(acetyl)-AMC. This suggests that the HDACs interact with BOC-Lys(crotonyl)-AMC, but potentially cannot release the compound (Fig. 8). Future studies will examine to what extent HDACs require the peptide context of histone H3 for efficient decrotonylation activity.

## Discussion

A key finding of our study is that histone crotonylation is surprisingly abundant in the intestinal epithelium, especially the crypt fraction of the small intestine and the colon. This supports the notion that histone crotonylation does not simply reflect 'metabolic noise', as in the ability of p300 to use crotonyl-CoA to modify histones, but has specific functions, which might vary between tissues. We also find high levels of histone crotonylation

**Fig. 2** H3K18cr ChIP-seq from colon epithelium analysis. ChIP-sequencing on isolated colon epithelial cells from two mice. **a** Browser view of a segment from chromosome 1 showing a representative profile of the distribution of H3K18cr peaks with relationship to genes. Relative enrichment of the combined replicate sets of ChIP and input in linear scale are shown, probes are 500 bp, 250 bp overlap. **b** Average distribution of ChIP-seq normalized read counts with relation to genes shows that histone H3K18cr is highly enriched over transcription start sites (TSS) in colon epithelial cells. **c** Link between H3K4me3 and H3K18cr, using MACS peak quantification and an aligned probe plot. Probes were ranked according to H3K4me3 signal strength and span 5 kbp around MACS peaks. **d** Average distribution of reads in linear scale with relation to genes' TSS, showing enrichment over these sites. **e** Relationship between H3K18cr enrichment over TSS and mRNA levels of the corresponding genes from cells isolated from the mouse colon epithelium were quantified using mRNA-seq (three biological replicates) and the normalized read counts over genes were divided into percentile bins as indicated, from lowly expressed genes (0–25 percentile) to very highly expressed genes (99–100 percentile). H3K18cr over TSS ±0.5 kbp of genes belonging to the expression bins was quantified and is shown in box-whisker plots. **f** KEGG pathway terms and their adjusted *p*-values of significance of genes with the highest 10 percentile H3K18cr associated (MACS) peaks. Only results with −log$_{10}$(p) > 6 are shown, see Supplementary Fig. 6 for all results. Cancer pathways are highlighted (red terms)

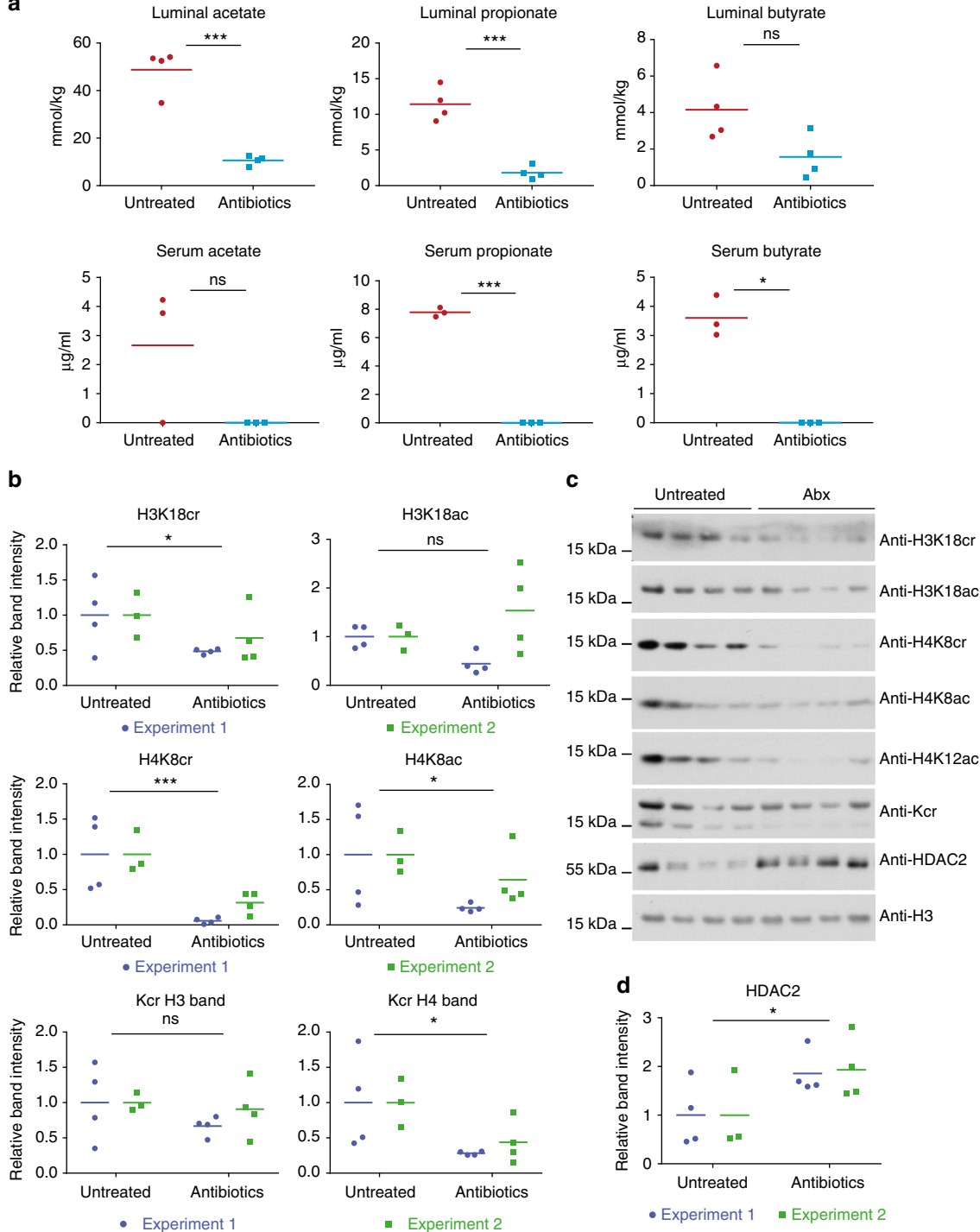

**Fig. 3** Microbiota depletion affects colonic histone crotonylation and HDAC2. Antibiotic treatment led to a decrease in luminal and serum SCFA levels in mice ($n \geq 3$, from experiment 2). **a** Acetate, propionate, and butyrate concentrations were measured in the colon lumen and serum by gas chromatography. Unpaired $t$-tests were conducted, *$p$-value < 0.05 and ***$p$-value < 0.001. Values of zero were below detectable levels. **b** Quantifications of western blot analysis of colon extracts from untreated and treated mice, $n \geq 3$. Experiments 1 and 2 are repeat experiments. Center values (small bar) are the average of the treatment group relative to the untreated group. Two-way ANOVA (two-tailed) was performed on quantified bands to compare the effect of treatment for both experiments together; * corresponds to a $p$-value of < 0.05 and *** corresponds to <0.001. The quantification showed a statistically significant decrease in H4 crotonylation as detected by the anti-Kcr antibody and in H4K8cr, H4K8ac, and H3K18cr levels upon antibiotics treatment. **c** Global changes in various colon histone crotonylation and acetylation marks and HDAC2 as seen in representative western blots of colon extracts, from experiment 1. **d** Two-way ANOVA was performed on quantified bands from western blotting analysis with anti-HDAC2. A statistically significant increase was observed ($p$-value < 0.05)

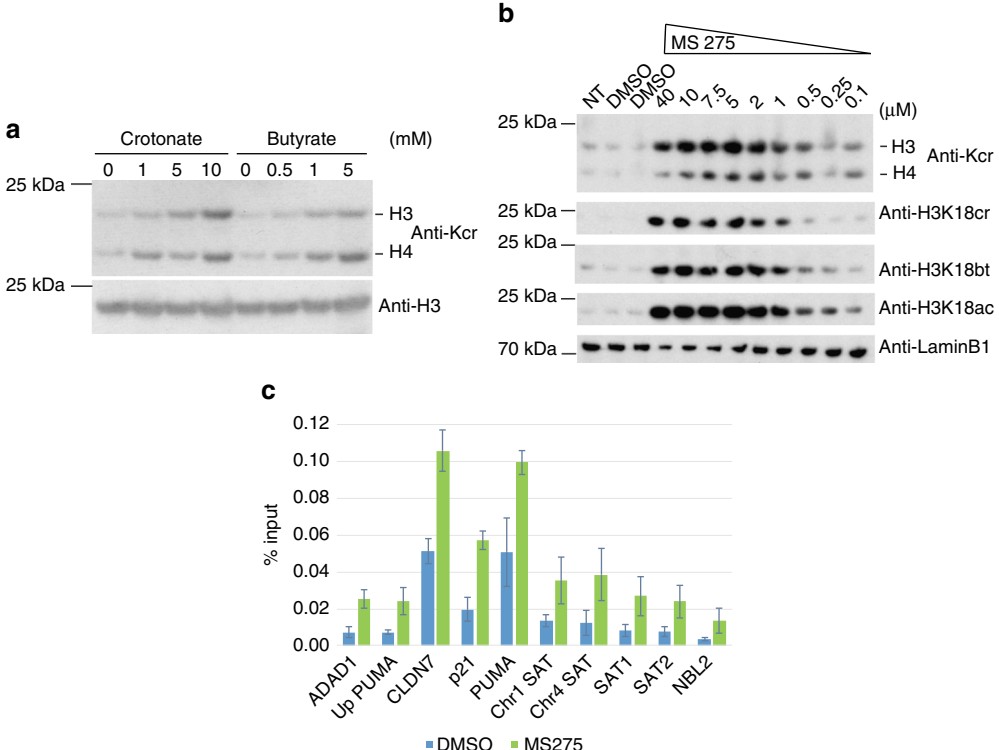

**Fig. 4** Butyrate and class I HDAC inhibition promote histone crotonylation. **a** Western blot analysis with indicated antibodies of whole cell extracts of small intestinal organoids treated for 48 h with indicated amounts of SCFAs. Representative western blot of two repeat experiments. **b** HCT116 cells were treated with MS275 or DMSO (vehicle) for 18 h, whole cell extracts collected, and analyzed by western blot using indicated antibodies; anti-Kcr: anti-crotonyl-lysine antibody, NT: not treated. **c** Increase in histone H3K18cr over promoters of indicated genes and repetitive, heterochromatic sites (alpha-satellite sequences, NBL2) upon MS275 treatment of HCT116 cells for 18 h. Summary of ChIP-qPCR data of three repeat experiments, error bars are SEM

in the brain. As SCFAs are taken up from the colon into the blood stream (as supported by our own data, Fig. 3a), SCFAs might be taken up by brain cells, where they may affect histone modifications[25]. Future studies will determine if there are brain-specific functions for histone crotonylation.

We found that the SCFA butyrate, a well-known HDAC inhibitor, promotes histone crotonylation in intestinal cell and organoid culture. This led us to test other HDAC inhibitors, which showed that class I specific HDAC inhibitors strongly promoted histone crotonylation at concentrations at which they affected histone acetylation and implicated class I HDACs in histone decrotonylation. Our in vitro assays of recombinant, purified class I HDAC1-3 show that they are decrotonylases and demonstrates that HDAC inhibitors, including butyrate, reduce decrotonylation. Furthermore, the low $K_m$ value of HDAC1 with crotonylated histones suggests that it can efficiently decrotonylate this relatively less abundant histone modification. Together, these findings strongly suggest that SCFAs such as butyrate promote histone crotonylation by inhibiting the decrotonylase activity of HDACs in colon epithelial cells. Future studies will address to what extent SCFAs also promote crotonylation by acting as substrates for the generation of intracellular crotonyl-CoA.

Our finding that class I specific HDAC inhibitors promoted global histone crotonylation led us to explore where this increased histone crotonylation (H3K18cr) occurs in the genome in cultured colon carcinoma cells. We show that the increase is not because of a greater number of crotonylation peaks but a diffuse spread of crotonylation genome-wide, including within repetitive regions. Therefore, while H3K18cr 'peaks' do exist over TSS, these findings suggest that pervasive crotonylation also occurs across the genome and that it is controlled by class I HDACs.

Additionally, we show that histone crotonylation is modulated during the cell cycle and that class I HDACs play a role in this regulation. In synchronized cells, we find an increase of histone crotonylation during S phase which is enhanced when cells are treated with the HDAC inhibitor MS275. Newly synthesized histones are acetylated on several specific lysine residues (e.g., K5 and K12 on histone H4; K14, K18 on histone H3) prior to their deposition onto nascent chromatin[26, 27], reviewed in ref. [28]. These acetylations are then globally removed following nucleosome assembly, which is important for the maintenance of repressive chromatin, such as pericentromeric heterochromatin[19, 29]. Class I HDACs are targeted to replicating chromatin and mediate this deacetylation[30, 31]. Therefore, the levels of these histone acetylations are modulated in a cell cycle and HDAC-dependent manner. In this study, we provide evidence that histone crotonylation marks behave similarly to the pre-deposition acetylation marks: they are low in G1 arrested cells and increase as cells progress through S phase and this modulation depends on class I HDACs. Therefore, histone crotonylation is not simply modulated through dilution with non-crotonylated histones through the cell cycle, but appears to be actively regulated by HDACs. Future studies will address if histone crotonylation is also mediated prior to histone deposition onto nascent chromatin and will address how HDACs and other factors regulate histone crotonylation concomitant with chromatin replication.

Importantly, consistent with our finding that microbiota-generated SCFAs such as butyrate promote histone crotonylation in vitro, we show that depletion of the microbiota led to a loss of histone crotonylation. Our data suggest that the levels of histone crotonylation in the gut reflect the generation of SCFAs by the microbiota. Given the important role of microbiota-generated SCFAs in modulating immunity and metabolism, our work

proposes that the signaling between the microbiota and chromatin might be mediated through histone crotonylation. Future work will explore further the roles of histone crotonylation in normal gut physiology including host–microbiome interaction, inflammation, and disease.

A surprising finding of our work is that depletion of the microbiota of mice with antibiotics not only led to a drop in luminal and serum SCFAs, but also an increased expression of HDAC2 in colon. A reduction in histone crotonylation is consistent with both changes. A loss of SCFAs upon microbiota

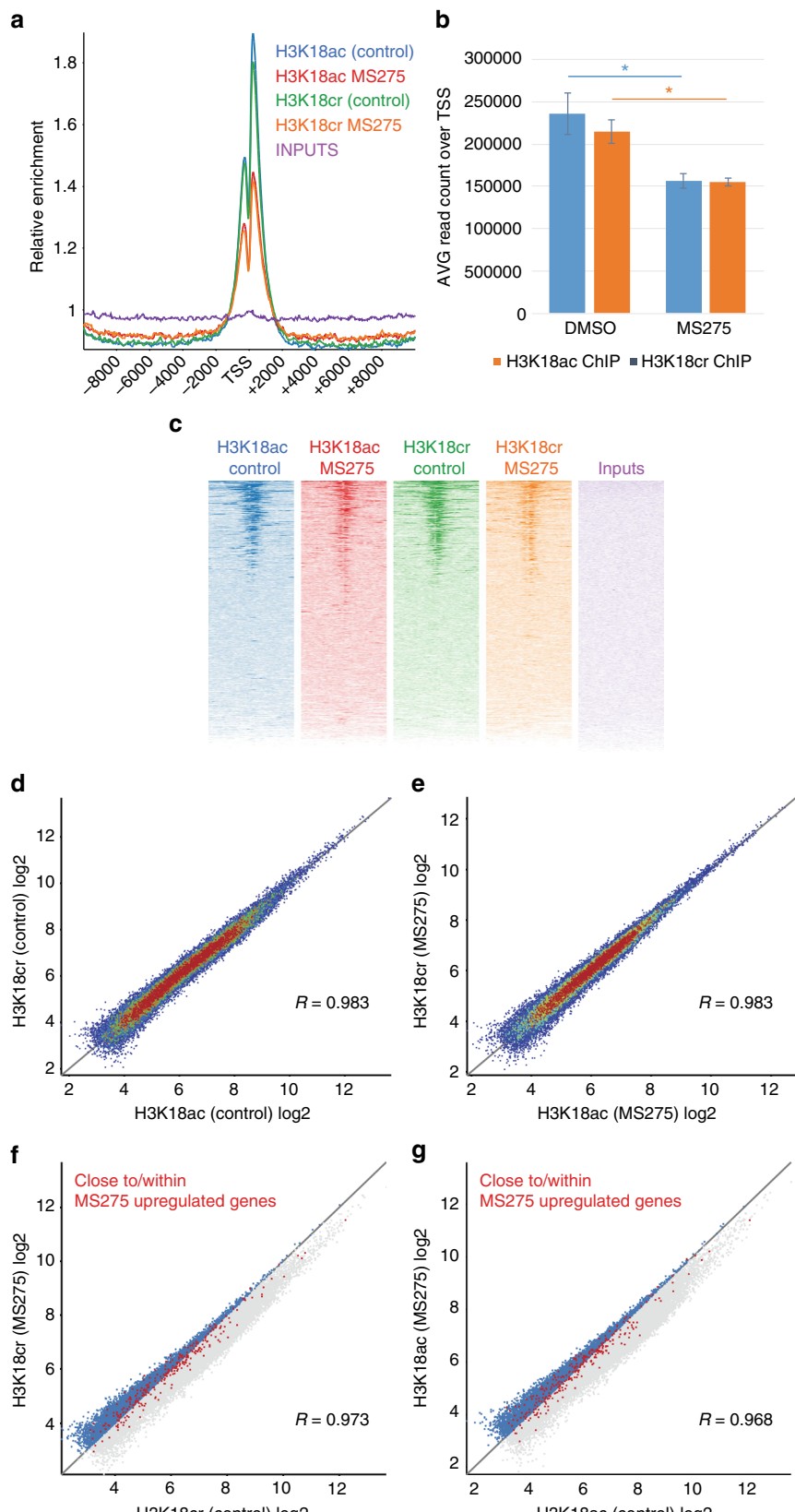

depletion may lead to stabilization or increased expression of HDAC2. In this context, it is interesting to note that a previous study has shown that the stability of HDAC2 is selectively reduced by the HDAC inhibitor valproic acid (a branched SCFA) or butyrate and this is mediated by Ubc8-RLIM targeted proteasomal degradation[32]. Remarkably, HDAC2 expression has been linked to colorectal tumorigenesis[33, 34]. It will be interesting to examine how this links to our finding that H3K18cr peaks occur over a significant number of genes linked to cancer pathways. Future studies will examine how HDAC2 expression, microbiota, and SCFAs are linked and how this affects histone crotonylation over specific genes and cancer progression in the colon.

While our study was in revision, a study by Yu, Wong, and collaborators also demonstrated histone decrotonylation activity of class I HDACs using transfection experiments in cell culture[35]. Our study complements and extends these and related findings[36, 37] as we explore the effect of HDAC inhibitors on the distribution of histone crotonylation, the link between histone crotonylation and the cell cycle, and investigate how the microbiota influence these processes. This expands our understanding of the functional spectrum of these exciting drug targets in cancer therapy and other diseases[38, 39] and provides a basis for the development of specific inhibitors of the decrotonylation versus deacetylation activity of these enzymes.

## Methods

**Mice**. Mice were C57BL/6 background, details available on request. Male mice were used for all experiments. All mice were kept in specific pathogen-free conditions and fed ad lib. Mice for the ChIP and RNA-seq experiments were housed at the Babraham Institute Biological Service Unit. All experimental protocols at Babraham Institute were approved by the Babraham Research Campus local ethical review committee and the Home Office (PPL 80/2488 and 70/8994). The antibiotics treatment experiments were performed at the University of Campinas. Male C57BL/6 mice were age 8–12 weeks were provided by the Multidisciplinary Centre for Biological Investigation (CEIMB) and all the experimental procedures were approved by the Ethics Committee on Animal Use of the Institute of Biology, University of Campinas (protocol number 3742-1). No sample size calculation was used as the minimum number of mice used was 3 or 4. Mice of the same age and breed were randomly put in the experimental groups. The order of samples from groups was mixed on collection. Sample size is reported in exact numbers and no samples are excluded from the analysis. No blinding was conducted.

**Antibiotic treatment of mice**. Mice received 200 μl of a mixture of antibiotics (5 mg/ml of neomicin, 5 mg/ml of gentamicin, 5 mg/ml of ampicillin, 5 mg/ml of metronidazole, and 2.5 mg/ml of vancomycin, Sigma Aldrich) daily for 3 days by gavage. The weight of the animals was monitored throughout the experiment. At the end of treatment period, feces were collected and snap-frozen in liquid nitrogen. After that, the animals were anesthetized using a mix of ketamine and xylazine (300 and 30 mg/kg, respectively) and the blood was collected by cardiac puncture. The blood was maintained at room temperature (RT) for 30 min and then centrifuged (3000×g, 8 min). The serum was collected and frozen at −80 °C. After euthanizing the animals by cervical dislocation, the entire intestine was harvested and the small intestine, colon, and cecum were isolated.

**Determination of fecal bacterial load**. Stool samples were collected on the third day after treatment by gavage with a mix of antibiotics or placebo. Fifty milligrams of the samples were used for extraction of the microbial genomic DNA using the Invitrogen™ PureLink™ Microbiome DNA Purification kit (Thermo Fisher

Scientific, MA, USA). Bacterial DNA was quantified by real-time PCR using primers complementary to 16S rDNA of Eubacteria (sense ACT CCT ACG GGA GGC AGC AGT; anti-sense ATT ACC GCG GCT GCT GGC)[40]. To determine the fecal bacterial load, a standard curve with serial dilutions was employed using genomic DNA extracted from *Escherichia coli* grown in vitro. Results obtained were normalized by the control condition (untreated mice).

**SCFA measurements**. Colonic luminal content samples were weighed into 1.5 ml tubes, crushed and homogenized in 100 μl of distilled water. Subsequently, 40 mg of sodium chloride, 20 mg of citric acid, 40 μl of 1 M hydrochloric acid, and 200 μl of butanol were added. The tubes were vortexed for 2 min and centrifuged at 18,000×g for 15 min. The supernatant was transferred to microtubes, and 1 μl was injected into the gas chromatograph. For serum measurements, 20 mg of sodium chloride, 10 mg of citric acid, 20 μl of 1 M hydrochloric acid, and 100 μl of butanol were added to 100 μl of serum samples. Tubes were vortexed and centrifuged as previously described and 1 μl was injected into the gas chromatograph. To quantify SCFAs, a calibration curve for the concentration range of 0.015–1 mg/ml was constructed. SCFAs measurements were performed following a recently published protocol[41]: chromatographic analyses were performed using an Agilent 6850 system with ExChrom software, equipped with a 7683B automatic liquid sampler, a flame ionization detector (FID) (Agilent Technologies, USA), and a fused-silica capillary RTX-WAX (Restec Corporation, U.S.) with dimensions of 60 m × 0.25 mm internal diameter (i.d.) coated with a 0.15-μm thick layer of polyethylene glycol. The initial oven temperature was 100 °C (hold 2 min), which was increased to 200 °C at a rate of 15 °C/min (hold 5 min). The FID temperature was maintained at 260 °C, and the flow rates of $H_2$, air, and the make-up gas $N_2$ were 35, 350, and 25 ml/min, respectively. Sample volumes of 1 μl were injected at 260 °C using a split ratio of approximately 25:1. Nitrogen was used as the carrier gas at 25 ml/min. The runtime for each analysis was 12.95 min.

**Small intestinal and colon epithelium extraction**. Animals were sacrificed by cervical dislocation or exposure to $CO_2$. Dissected small intestines and colons were opened longitudinally and washed three times with ice cold Hank's balanced salt solution without $Ca^{2+}/Mg^{2+}$ (HBSS). Intestinal and colon epithelium were dissociated with 30 mM EDTA/HBSS on ice with shaking for 30 min for small intestine and 1 h for colon; 50 ml Falcon tubes containing the tissue were then shaken vigorously by hand (2–3 shakes/second) for 5 min. The colon epithelium was incubated for an additional 10 min on ice in 30 mM EDTA/HBSS and further shaken for 5 min. Mucus and sub-mucosa were removed by dripping the material through a 100 μm followed by a 70 μm cell strainer. The extracted cells were pelleted at 475×g at 4 °C for 10 min. The cells were washed with ice cold HBSS and re-pelleted as above for further use. For the H3K18cr ChIP, three colons were combined.

**Purification and enzymatic digestion of histones**. For isolating a crypt-enriched fraction, villi were removed by shaking the small intestine pieces for 10 s after incubation for 5 min on ice in HBSS–30 mM EDTA and before the additional incubation in HBSS–30 mM EDTA for 15 min. Histones were then acid extracted following a protocol published in ref. [42]: one small intestine/colon was used per extraction followed by MS analysis. Small intestine epithelium, colon epithelium, and a crypt-enriched fraction were homogenized in lysis buffer (10% sucrose; 0.5 mM EGTA, pH 8.0; 15 mM NaCl; 60 mM KCl; 15 mM HEPES; 0.5% Triton; 0.5 mM PMSF; 1 mM DTT; 5 mM NaF; 5 mM $Na_3VO_4$; 5 mM Na-butyrate, cocktail of protease inhibitors (Sigma)). Nuclei were separated from the cytoplasm by centrifugation on sucrose cushions, washed in cold PBS, and then extracted in 0.4 N HCl overnight (o.n.) at 4 °C. Core histones, together with linker histones protein, were dialyzed against 100 mM ice-cold acetic acid. The concentration of purified samples was measured using the Bradford protein assay. Approximately 10 μg were separated on SDS-PAGE and bands corresponding to the histones H3 were excised and in-gel digested[43]. Briefly, gel bands were cut in pieces and destained with repeated washes in 50% acetonitrile (ACN) in $H_2O$, alternated with dehydration steps in 100% ACN. Gel pieces were then in-gel chemically alkylated by incubation with D6-acetic anhydride (Sigma 175641) in 1 M $NH_4HCO_3$ and $CH_3COONa$ solution as catalyzer. After 3 h at 37 °C with high shaking in a thermomixer, chemically modified gel slices were washed with $NH_4HCO_3$, alternated with ACN

**Fig. 5** H3K18cr and H3K18ac ChIP-seq on MS275-treated HCT116 cells. **a** Probe trendplot over TSS (±10 kbp) of reads from H3K18ac and H3K18cr ChIP-seq on HCT116 cells with and without MS275 treatment; **b** H3K18cr and H3K18ac ChIP-seq analysis shows a relative decrease in these marks over TSS upon MS275 treatment, error bars are SEM, n = 3, p < 0.05, paired t-test; **c** aligned probe plots over TSS (±5 kbp) of reads from H3K18ac and H3K18cr ChIP-seq with and without MS275 treatment, aligned probes were ranked according to read counts in the H3K18cr/MS275 ChIP-seq. **d** Scatterplot of read counts of H3K18ac versus H3K18cr MACS peaks of control (vehicle treated) cells and **e** of H3K18ac versus H3K18cr MACS peaks of MS275-treated cells. **f** Read counts in H3K18cr MACS peaks from control cells versus MS275-treated cells. MACS peaks close (+2 kbp) and within upregulated genes are in red. **g** Read counts in H3K18ac MACS peaks from control cells versus H3K18ac, MACS peaks of MS275-treated cells, MACS peaks close (+2 kbp) and within upregulated genes are in red. MACS peaks that show an increase in H3K18cr (**f**) or H3K18ac (**g**) on MS275 treatment are in blue. For both H3K18cr and H3K18ac, there is a disproportionate larger number of MACS peaks linked to MS275-upregulated genes that also show an increase in H3K18cr (**f**) or H3K18ac (**g**) on MS275 treatment compared to those that show a decrease in these modifications (p < 0.0001, $\chi^2$ test)

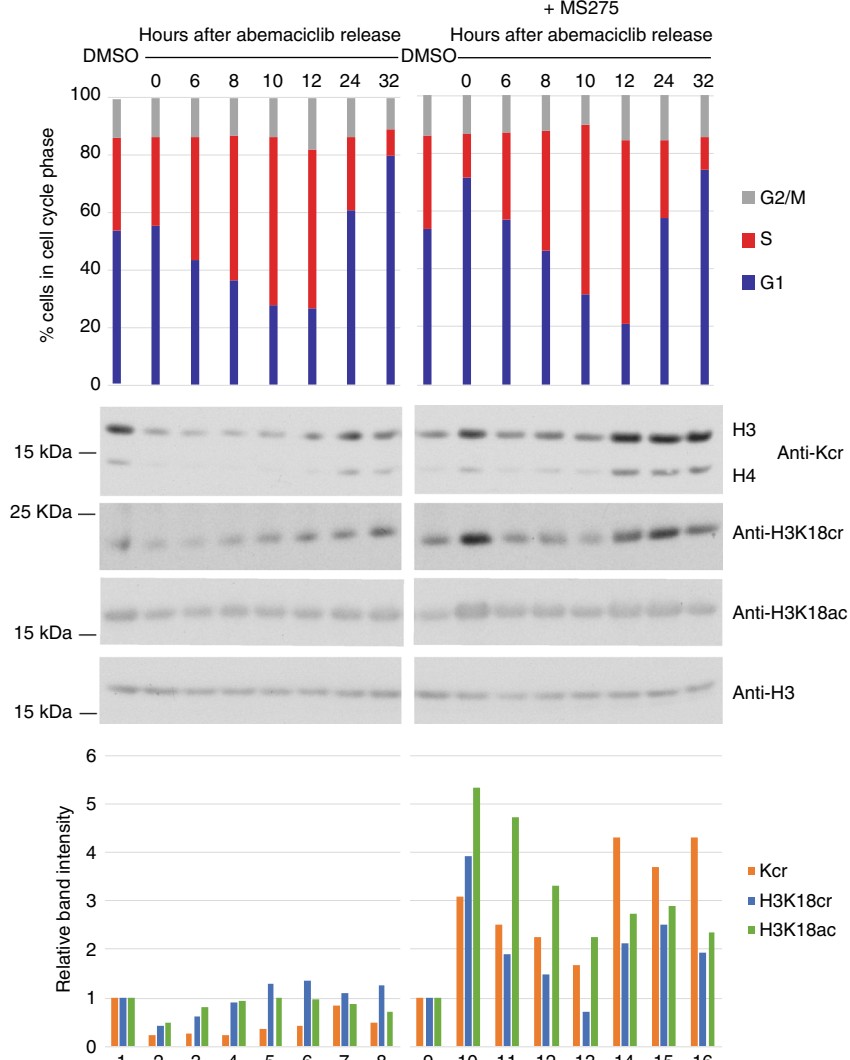

**Fig. 6** Histone crotonylation is cell cycle regulated by class I HDACs. Cell cycle block and release experiment on HCT116 cells using CDK4/6 inhibitor abemaciclib with and without MS275. Lanes 1 and 9: asynchronous cells, lane 2: G1 arrested cells, lanes 3–8: increase in histone crotonylation (Kcr), H3K18cr and H3K18ac upon release into S phase. Lanes 10–16: histone crotonylation, H3K18cr and H3K18ac are upregulated during a G1 arrest and S phase when class I HDACs are inhibited with MS275. For the experiments in lanes 9–16, cells were blocked in G1 using 15 nM abemaciclib in the presence of 5 μM MS275 and released into S phase in the presence of 1 μM MS275. Cell cycle profiles are shown at the top, western blots in the middle, and quantifications of those, as calculated relative to H3 and normalized to the DMSO (vehicle) sample, at the bottom. Representative of two experiments is shown

at increasing percent (from 50 to 100%). In-gel digestion was performed with 100 ng/μl trypsin (Promega V5113) in 50 mM NH$_4$HCO$_3$ at 37 °C overnight. Chemical acetylation occurs on unmodified and monomethylated lysines and prevents trypsin digestion at these residues, thus producing a pattern of digestion similar to that obtained with the Arg-C protease (the so-called "Arg-C like" in-gel digestion pattern). The resulting histone peptides display an optimal length for MS detection and enhanced hydrophobicity that increases their separation at ultra-pressure chromatographic regimes.

Finally, digested peptides were collected and extracted using 5% formic acid alternated with ACN 100%. Digested peptides were desalted and concentrated using a combination of reverse-phase C18/C "sandwich" system and strong cation exchange (SCX) chromatography on hand-made micro-columns (StageTips[44, 45]). Eluted peptides were lyophilized, suspended in 1% TFA in H$_2$O, and then subjected to LC-MS/MS.

**LC-MS/MS**. The peptide mixtures were analyzed by online nano-flow LC-MS/MS using an EASY-nLC 1000 (Thermo Fisher Scientic) connected to a QExactive (Thermo Fisher Scientific) through a nano-electrospray ion source. The nano-LC system was operated in one column setup with a 25-cm analytical column (75 μm inner diameter, 350 μm outer diameter) packed with C18 resin (ReproSil, Pur C18AQ 1.9 m, Dr. Maisch, Germany) configuration. Solvent A was 0.1% formic acid (FA) in ddH$_2$O and solvent B was 80% ACN with 0.1% FA. Samples were injected in an aqueous 1% TFA solution at a flow rate of 500 nl/min. Peptides were

separated with a gradient of 0–40% solvent B for 100 min, followed by a gradient of 40–60% in 5 min, and 60–95% over 5 min at a flow rate of 250 nl/min. The Q-Exactive instrument was operated in the data-dependent acquisition (DDA) to automatically switch between full scan MS and MS/MS acquisition. Survey full scan MS spectra (from $m/z$ 300–1150) were analyzed in the Orbitrap detector with resolution $R = 60,000$ at $m/z$ 200. The 10 most intense peptides were sequentially isolated to a target value of $3 \times 10^6$ and fragmented by high-energy collisional dissociation (HCD) with a normalized collision energy setting of 27%. The maximum allowed ion accumulation times were 20 ms for full scans and 50 ms for MS/MS and the target value for MS/MS was set to $1 \times 10^6$. Standard mass spectrometric conditions for all experiments were as follows: spray voltage, 2.4 kV; no sheath and auxiliary gas flow.

**MS data analysis and relative abundance profiling**. Acquired RAW data were analyzed by MaxQuant(MQ) software v1.5.2.8, using the Andromeda search engine[46]. Uniprot Mouse database (70,902 entries) was used for peptide identification. Enzyme specificity was set to Arg-C. Estimated false discovery rate of all peptide identifications was set at a maximum of 1% (Decoy database-based approach). Mass tolerance for searches was set to a maximum of 6 parts per million (ppm) for peptide masses and 20 ppm for HCD fragment ion masses. A maximum of three missed cleavages was allowed. In the search, we focused on lysine methylation and acylation, including as variable modifications: D3-acetylation (+45.0294 Da), D3-acetylation (+45.0294 Da) plus monomethylation (+14.016 Da),

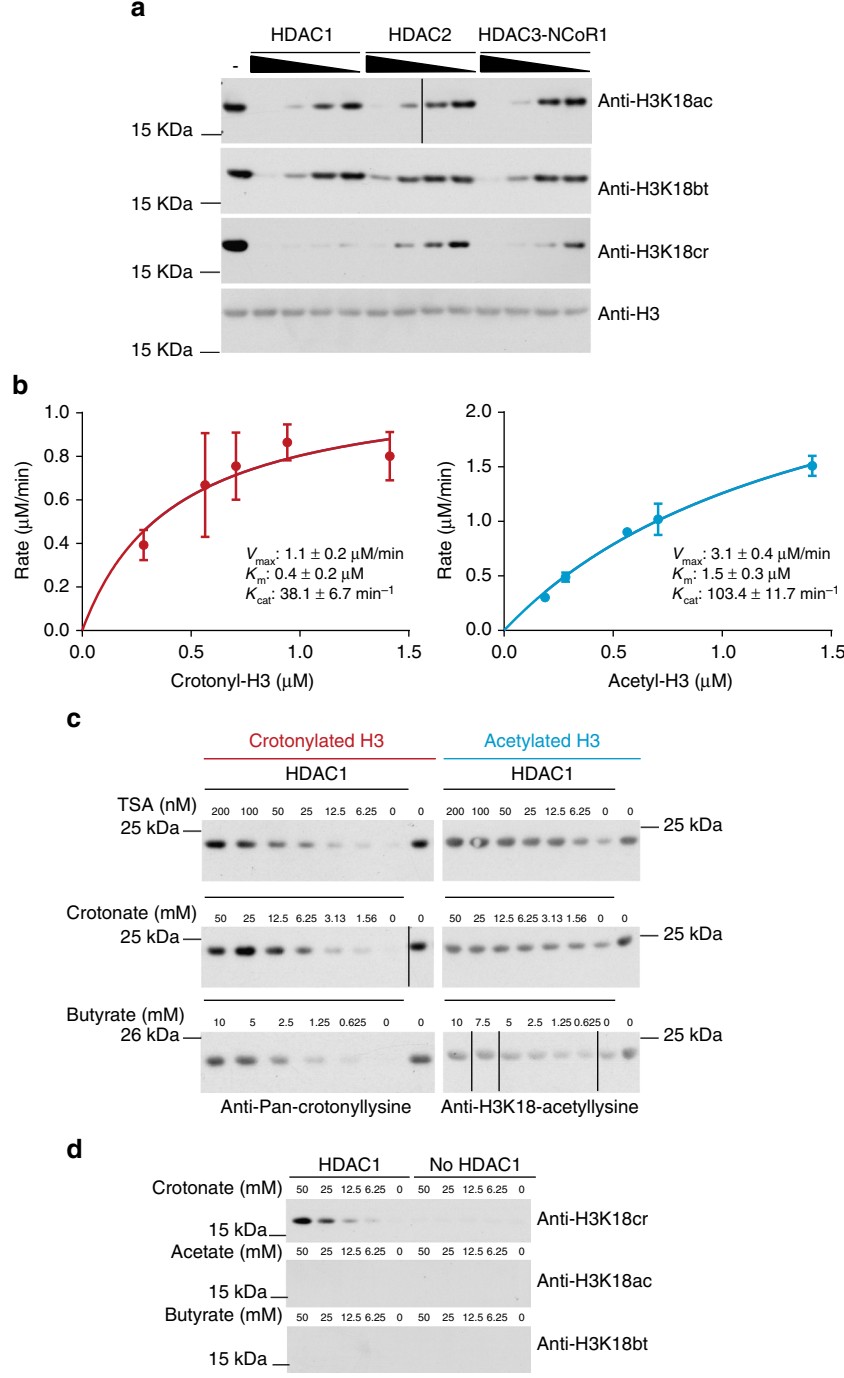

**Fig. 7** Class I HDACs are histone decrotonylases. **a** Histone H3 decrotonylation and deacetylation in vitro by HDAC1, HDAC2, or HDAC3/Ncor1 complex; 5.65 μM histones were crotonylated or acetylated in vitro and then subjected to removal of the modification by the indicated HDACs. HDAC1 was 0.25, 0.12, 0.06, and 0.03 μM. HDAC2 was 0.18, 0.09, 0.05, and 0.02 μM. HDAC3/Ncor1 complex was 0.45, 0.23, 0.11, and 0.06 μM. **b** Comparative kinetics of HDAC1 decrotonylation and deacetylation; 5.65 μM histones were crotonylated or acetylated and then subjected to removal of the modification by 0.03 μM HDAC1 for different lengths of time. Samples were analyzed by dot blotting and initial rates of reaction were determined by plotting substrate removal over time. Kinetic parameters $V_{max}$, $K_m$, and $K_{cat}$, error bars are SEM, $n = 3$. **c** Effect of HDAC inhibitors TSA, crotonate, and butyrate on deacetylation and decrotonylation by HDAC1 in vitro. Representative blots of two repeat experiments are shown. **d** Histone crotonylation by HDAC1 using crotonate in vitro. Incubation of crotonate, acetate, or butyrate with or without HDAC1 followed by western blotting analysis with anti-H3K18ac/bt/cr. Western blot of HDAC1 and crotonate assay is representative of two western blots

dimethylation (+28.031 Da), trimethylation (+42.046 Da), acetylation (+42.010 Da), and crotonylation (+68.074 Da). The use of high-accuracy criteria for HCD fragment ions tolerance (20 ppm) guarantee the capability to discriminate among other possible forms of acylation (e.g., lysine butyrylation (70.0418 Da) and β-hydroxybutyrylation (86.0367 Da)), therefore they were not included in the search. MaxQuant search results were exported and peptides with Andromeda score <60 and localization probability score <0.75 were removed. Filtered data were subjected

to manual inspection and validation using the viewer.exe module integrated in MQ software[47]. Extracted ion chromatograms were constructed for each precursor based on the $m/z$ value, using a mass tolerance of 10 ppm with a mass precision up to four decimals. For each histone-modified peptide, the relative abundance percentage (RA%) was estimated by dividing the area under the curve (AUC) of each modified peptide over the sum of the areas corresponding to all observed isoforms of that peptide, including the unmodified forms[48]. Significant changes among crypt

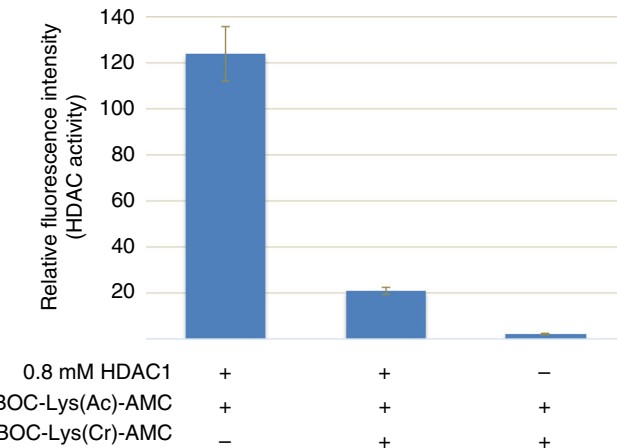

| | | | |
|---|---|---|---|
| 0.8 mM HDAC1 | + | + | − |
| BOC-Lys(Ac)-AMC | + | + | + |
| BOC-Lys(Cr)-AMC | − | + | + |

**Fig. 8** BOC-Lys(crotonyl)-AMC inhibits deacetylation by HDAC1. A fluorometric in vitro assay showing that HDAC1 efficiently deacetylates the BOC-Lys(acetyl)-AMC substrate alone, but not in the presence of same amounts of BOC-Lys(crotonyl)-AMC. Performed in triplicate, error bars are standard deviation

and colon versus small intestine fraction have been calculated with a two-way ANOVA test using the Perseus software[49]. p-Value < 0.01 (1% FDR) were considered as significant.

**Antibodies and western blot analysis**. Anti-lysine crotonyl antibody (PTM-501, 1:5000), anti-crotonyl-histone H3 lys18 (anti-H3K18cr, PTM-517, 1:5000), and anti-butyryl-histone H3 lys18 (anti-H3K18bt, PTM-306, 1:5000) were from PTM BIOLABS; anti-tri-methyl H3 lys4 antibody was from Active Motif (anti-H3K4me3, Cat.39159); anti-crotonyl-histone H4 lys8 antibody (anti-H4K8cr, ab201075), anti-acetyl-histone H3 lys18 (anti-H3K18ac, ab1191, 1:10000), anti-histone H4 (ab31827, 1:40000) anti-histone H3 (ab1791, 1:40000), and anti-LaminB1 (ab16048, 1:5000) were from Abcam; mouse monoclonal anti-HDAC1 (Clone 2E10, 1:5000) from Millipore, monoclonal anti-HDAC2 (C-8) (sc9959, 1:5000) from SantaCruz, and mouse polyclonal HDAC3 (BD61124, 1:5000) from BD biosciences.

Western blots were washed with tris-borate-sodium-0.05% Tween-20 (TBS-T) and developed using enhanced chemiluminescence (ECL); uncropped western blots are shown in Supplementary Figures 12–14.

**Cryosections and immunofluorescence staining**. Cryosections were prepared from adult murine tissue, fixed for 90 min at RT with 4% formaldehyde in phosphate buffered saline (PBS). After 2 × 5 min washes with 1× PBS, samples were incubated in 30% sucrose overnight at 4 °C and embedded in Cryomatrix (ThermoFisher 6769006). Frozen blocks were cut with the Leica CM1860 cryotome to 8 µm sections and attached to Superfrost Plus slides (ThermoFisher 4951PLUS4). Before staining, sections were brought to RT, dried for 3 min at 60 °C, permeabilized for 20 min at RT in 1% Triton-X-100/PBS, and unmasked for 30 min at 95 °C in a citrate based-unmasking buffer (VECTOR H3300) followed by 3 × 5 min washes in PBS. Slides were blocked for 1 h in 5% FBS/PBS (FBS: fetal bovine serum) at RT followed by primary antibody (pan-crotonyl PTM Biolabs Inc. #PTM501, 1 µg/ml final conc.) incubation in blocking solution for 1 h at RT. After 4 × 5 min washes in 1× PBS, secondary antibody (AlexaFluor 488 Invitrogen A11008, 1 µg/ml final conc.) was applied for 1 h at RT in blocking solution supplemented with DAPI. After 4 × 5 min washes in 1× PBS, samples were mounted in Vectashield H-1000 and sealed. Controls without the primary antibody were processed accordingly. Imaging was performed with the Zeiss780 confocal microscope using 20× air and 63× oil immersion objectives at optimal resolution settings. Z-stacks of whole sections were imaged and further processed to maximum projections with ImageJ software. For optimal print results background correction and contrast enhancement with up to 3% pixel saturation were performed with ImageJ.

**Cell culture and cell cycle analysis**. Human colon carcinoma cells (HCT116) were a gift from Simon Cook's lab (Babraham Institute) who obtained them from Bert Vogelstein, John Hopkins University, Baltimore. This cell line is not in the database of commonly misidentified cell lines (ICLAC). They were grown in DMEM media containing glucose and pyruvate, 10% FBS, 2 mM L-glutamine, 100 units/ml penicillin, and 100 µg/ml streptomycin. For the cell cycle analysis, HCT116 cells were blocked at G1 with 15 nM of abemaciclib (LY2835219, Seleck Chemicals) for 48 h and released by washing 2× with PBS and adding fresh medium. Class I HDAC inhibition was with 5 µM MS275 for 48 h. Cells were

washed 2× with PBS and fresh medium supplemented with 1 µM MS275 was added. Cells were harvested at indicated intervals after release and one half was analyzed by western blotting. The remainder were used to determine cell cycle profiles with propidium iodide staining using the BD Pharmingen PI/RNAse staining buffer on LSRII Flow Cytometer (BD Biosciences) and Cell Cycle tool on FlowJo 10.0.8. HDAC1 over-expression was performed by transfecting HCT116 cells using Lipofectamine® 2000 Transfection Reagent following manufacturer's protocol using 50 µL of reagent and 12 µg of p181 pK7-HDAC1 (GFP) plasmid DNA (gift from Ramesh Shivdasani, Addgene plasmid # 11054[50]) or an N-terminal deletion mutant in a 60-mm dish for 16 h.

**Whole cell extract preparations**. For whole cell extract preparation, cells were detached with trypsin, washed in PBS, and boiled in Laemmli sample buffer (Biorad) for 5 min. The extracts were briefly sonicated to remove high molecular weight DNA before loading on an SDS-polyacrylamide gel for electrophoresis.

**Intestinal organoid seeding and cultures**. Small intestinal crypts were derived from wildtype C57BL/6 mice using a slightly modified protocol from reference[51]. In brief, collected small intestines were opened longitudinally and the majority of villi removed by gentle scraping with a coverslip. The tissue was cut to 3–5 mm pieces, washed five times with cold PBS, and vigorous shaking. After 30 min incubation on ice with 2 mM EDTA/ PBS, the remaining villi were removed with short shaking and tissue pieces were incubated for additional 30 min in 5 mM EDTA/PBS on ice. After another short shake, crypts were passed through a 40-µm cell strainer and pelleted at 425×g 1500 rpm at 4 °C for 10 min. The pellet was washed with cold PBS and 100–200 crypts were suspended in 50 µl Red-phenol-free Matrigel (BD Biosciences) droplets. After polymerization, complete medium containing advanced DMEM/F12 (Sigma), 2 mM Glutamax (Invitrogen), 10 mM HEPES (Gibco), 100 U/ml penicillin/streptomycin (Invitrogen), 1 mM N-acetyl-cysteine (Sigma), 1× B27 supplement (Invitrogen), 1× N2 supplement (Invitrogen), 50 ng/ml mouse EGF (Peprotech), 100 ng/ml mouse Noggin (Peprotech), and 10% human R-spondin-1-conditioned medium from R-spondin-1-transfected HEK293T cells (Cultrex) was added to the cultures. Medium was changed every 3 days and organoids passaged after 7–10 days.

**ChIP-seq of extracted colon epithelium**. The colon epithelium cell pellet was resuspended in 10 ml of PBS-1% formaldehyde and fixation was carried out for 10 min, at RT with gentle agitation. The reaction was quenched by the addition of glycine (0.125 M final concentration) and the cells were pelleted at 475×g at 4 °C for 10 min, washed once with PBS, re-pelleted and either snap-frozen in liquid nitrogen for storage at −80 °C or processed further immediately. The cell pellet was resuspended and incubated for 10 min in 500 µl 50 mM Tris-HCl, pH 8.0, 10 mM EDTA 1% SDS (ChIP lysis buffer) on ice. Sonication of the chromatin to 100–500 bp fragment size range in polystyrene tubes was performed with a water-cooled Bioruptor (Diagenode), high power, 4 °C, 12 cycles, 30 s on, 30 s off. The sonicated material was transferred to a 1.5-ml tube, incubated for 30–45 min on ice, and pelleted at 20,800×g for 10 min, 4 °C to precipitate SDS. We pooled three colons to perform three ChIP experiments, using 20–25 µg equivalent of DNA for each experiment. For immunoprecipitation, the chromatin was diluted 1:10 with ChIP dilution buffer (16.7 mM Tris-HCl, pH 8.0, 1.2 mM EDTA, 167 mM NaCl, 1.1% Triton-X-100), 5 µg anti-H3K18cr antibody (PTM-517) or H3K4me3 antibody (Active Motif Cat.39159) per 20 µg DNA was added and this was incubated overnight on a rotating wheel at 4 °C. One percent input chromatin was collected and kept on ice. Immunoprecipitation of chromatin complexes was with Protein A-coated Dynabeads (Novex, Cat.10001D); 20–30 µl of bead suspension were washed two times with ChIP dilution buffer and the antibody–chromatin mix was added to the beads. Immunoprecipitation was for 2 h at 4 °C on a rotating wheel. Following this incubation, tubes were spun briefly and bound material was separated from unbound using a magnetic stand on ice.

All washes were performed at 4 °C for 5 min on a rotating wheel using 20× volumes with respect to the beads volume used. Beads were washed 1× with low-salt buffer (20 mM Tris-HCl, pH 8.0, 2 mM EDTA, 150 mM NaCl, 1% Triton-X-100, 0.1 % SDS), 2× with high-salt wash buffer (20 mM Tris-HCl, pH 8.0, 2 mM EDTA, 500 mM NaCl, 1% Triton-X-100, 0.1 % SDS), and 1× with 10 mM Tris-HCl, 1 mM EDTA (1× TE). Elution of DNA from beads was with 200 µl of freshly prepared elution buffer (0.1 M NaHCO3, 1% SDS) at 65 °C for 30 min in a thermomixer at 1000 rpm. Supernatant was separated using a magnetic stand and transferred to a fresh tube. After bringing all inputs to 200 µl with elution buffer, both chromatin and input samples were reverse cross-linked by adding 8 µl of 5 M NaCl followed by an incubation at 65 °C overnight at 300 rpm in a thermomixer. Proteinase K Solution (Ambion, Cat:AM2548) was added to samples to a final concentration of 0.25 mg/ml and incubated for 2 h at 65 °C, 300 rpm. Chipped DNA was purified with QIAquick PCR purification kit (Qiagen, Cat.28104) and quantified using Qubit™ 3.0 fluorimeter. Library preparation was performed from 5 ng of purified DNA using the NEBNext® Ultra™ II DNA Library Prep Kit for Illumina® with the following modifications: Illumina Tru-Seq adaptors were used and library amplification was performed with the KAPA PCR Amplification kit (KAPA, Cat. KK2501) using 11 cycles. Libraries were sequenced on a HiSeq2500 sequencer (Illumina) according to manufacturer's instructions.

**Table 2 Primers used in ChIP-qPCR**

| Name | Feature | Forward (5′→3′) | Reverse (5′→3′) |
|---|---|---|---|
| up PUMA | Intergenic | GTTGCCAGTTACCACACCCT | CCCAACTGTCCTTGCTGCTA |
| ADAD1 | TSS of not expressed gene | GCTTCAGGACGTGTGAGGTA | TACCTGCGTGAGGGTTGTTT |
| CLDN7 | TSS of expressed gene | ACTCTAAGGGAGGGGAACGAT | CCACTGGGACCTAAAGCCG |
| p21 | TSS of expressed gene | AGCAGGCTGTGGCTCTGATT | CAAAATAGCCACCAGCCTCTTCT |
| PUMA | TSS of expressed gene | GTGTGTGTGTCCGACTGTCCCG | AAGGAGGACCCAGGCGCTGT |
| Chr1 SAT | Repetitive region | TCATTCCCACAAACTGCGTTG | TCCAACGAAGGCCACAAGA |
| Chr4 SAT | Repetitive region | CTGCACTACCTGAAGAGGAC | GATGGTTCAACACTCTTACA |
| SAT1 | Repetitive region | GAACCTGTGTTGCTGCTTTG | TTCAAAGGTACTCTGCTTGGTACA |
| SAT2 | Repetitive region | TGAATGGAATCGTCATCGAA | CCATTCGATAATTCCGCTTG |
| NBL2 | Repetitive region | TCCCACAGCAGTTGGTGTTA | TTGGCAGAAACCTCTTTGCT |

**ChIP-seq and ChIP-qPCR of HCT116 cell extracts**. HCT116 cells were treated with either 5 µM MS275 or DMSO for 18 h. Cells were trypsinized and fixed as indicated in the ChIP-seq section above. Fixed cells were re-suspended in Sonication buffer (150 mM NaCl, 25 mM Tris pH 7.4, 5 mM EDTA, 0.1% Triton, 1% SDS complemented with 10 mM sodium butyrate and protease inhibitor cocktail (P8340, Sigma)) and sonicated as described above for 15 cycles. After centrifugation at 14,000×g for 10 min, supernatant was diluted 10× in ChIP dilution buffer; 1% input chromatin was collected and kept on ice and 30 µg equivalent of DNA per sample was incubated overnight on a rotating wheel at 4 °C with 5 µg of anti-H3K18cr antibody (PTM-517, PTM Biolabs) or 0.3 µg of anti-H3K18ac antibody (ab1191, Abcam). Twenty microliters of Magnetic ProtA/G Beads (Millipore) were added to the samples and incubated on a rotating wheel for 3 h at 4 °C. Antibody-bound beads were washed as described above. Chip DNA was eluted at 65 °C for 30 min in 200 µl of elution buffer. De-crosslinking and DNA elution of both ChIP and input samples was performed as described in the ChIP-seq section. Real time qPCR analysis was carried out on input and ChIP DNA samples using the SYBR® Green PCR Master Mix (Applied Biosystems) and run on a BioRad CFX96 qPCR system. Each experiment has been carried out two times (biological replicates) and each sample has been run in triplicate (technical replicate). One percent of starting chromatin was used as input and data were analyzed accordingly. Primers were used at a final concentration of 250 nM with 62 °C as annealing/extension temperature and are listed in Table 2.

**RNA-seq**. RNA was extracted from both HCT116 cells (four biological replicates) and mouse colon epithelium (three mice as biological replicates) using the RNeasy Plus Mini Kit (Qiagen), following the manufacturer's instructions. Extracted RNA was quantified with Nanodrop and the quality assessed on a Bioanalyzer (Agilent). Library preparation was performed from 500 ng of RNA using the NEB Next® Ultra™ Directional RNA Library Prep Kit for Illumina® and the NEBNext® Poly (A) mRNA magnetic isolation module. Illumina Tru-Seq adaptors were used and library amplification was performed with the KAPA PCR Amplification kit (KAPA, Cat. KK2501) using 14 cycles. Libraries were sequenced on a HiSeq2500 sequencer (Illumina) according to the manufacturer's instructions.

**Bioinformatic analysis**. *RNA-seq:* Sequencing reads were adaptor trimmed using Trim Galore! (version 0.4.2) and mapped to the mouse (GRCm38/mm10) reference genome with HiSat2 (version 2.0.5). Analysis of RNA-seq data was performed with SEQMONK version 1.36.0 on filtered reads with a MAPQ of >60 for uniquely mapped reads. Read counts were quantified using the RNA-seq quantitation pipeline implemented in Seqmonk, quantifying only probes with at least one read. Probe read values were corrected for transcript length and divided into percentile bins according to their average expression levels of three replicates.

*ChIP-seq analysis:* For mouse colon and HCT116 H3K18cr, two replicates and for H3K4me3 three biological replicates were used. Sequencing reads were adaptor trimmed using Trim Galore! (version 0.3.8 and 0.4.2) and mapped to the mouse (GRCm38/mm10) or human (GRCh38) reference genomes with Bowtie2 (version 2.0.4.). Analysis of ChIP-seq data was performed with SEQMONK version 1.36.0 on filtered reads with a MAPQ score of >42, duplicate reads were always removed. For Fig. 2b, the genome was segmented in 1000 bp non-overlapping 'probes' and read counts were quantified for each probe and normalized to the largest datastore. For Fig. 2c, H3K4me3 peaks were identified using the MACS peak finder embedded in the SEQMONK program with the ChIP-seq data of H3K4me3 from colon epithelial cells and INPUT as reference. Selected fragment size was 300 bp and *p*-value significance threshold was $10^{-5}$. Reads were quantitated in and ±5 kbp around the MACS peaks. For Fig. 2d, read counts were quantified over the TSS (using a window of ±1 kb upstream of genes) and probes that had more than 100 reads in input were removed in both input and ChIP. For Fig. 2e, read counts were quantified over the TSS (using a window of ±0.5 kb upstream of transcripts) and were overlapped with RNA-seq data. KEGG pathway analysis was with DAVID 6.8 with electronic annotations excluded.

**ChIP-qPCR data analysis**. Prior to analysis, a logarithmic transformation of the data was carried out. Subsequently, we carried out a two-way ANOVA test followed by Holm-Sidak's multiple comparison test. Data were analyzed using the Graphpad Prism Software.

**In vitro enzymatic assays of histone decrotonylation and deacetylation**. Recombinant human histone H3.1 (NEB) was acetylated or crotonylated in vitro using the recombinant catalytic domain of p300 (human, ENZO) in presence of acetyl-CoA or crotonyl-CoA (Sigma-Aldrich), respectively. The reaction was carried out with 5.65 µM histone H3.1 and 0.66 µM P300 catalytic domain in 50 mM Tris-HCL pH 8, 50 mM KCl, 0.1 mM EDTA, 0.01 % Tween-20, 10% glycerol, 1 mM DTT, and 87 µM crotonyl-CoA or acetyl-CoA at 30 °C for 2 h. The reaction was stopped by heating at 65 °C for 5 min and the histones were diluted with two volumes of HDAC buffer (25 mM Tris-HCl, pH 7.5, 50 mM KCl, 1 mM MgCl₂, 1 µM ZnSO₄) for either decrotonylation or deacetylation reactions. Decrotonylation/deacetylation was typically performed with 1.75 µM modified histones and 0.12 µM HDAC1 (recombinant, human, Active Motif) or 0.18 µM HDAC2 (recombinant, human, ABCAM) at 30 °C for 2 h. Recombinant HDAC3/Ncor1 complex was from ENZO Life Sciences. All HDACs were produced in insect cells and purified. After the reaction, the histone modifications were identified by western blot. The pixel intensities of the western blot bands were quantified with the Image J software.

For the enzyme kinetic analysis, histone H3 was acetylated or crotonylated in vitro as described above. Dot blot western analysis with synthetic H3 peptides that were specifically crotonylated or acetylated at K18 indicated full crotonylation and acetylation of the histone H3 at K18 under these conditions, therefore, we assume that the histone had been fully crotonylated or acetylated. Different dilutions of modified histones were prepared in HDAC buffer (25 mM Tris-HCl, pH 7.5, 50 mM KCl, 1 mM MgCl₂, 1 µM ZnSO₄) for either decrotonylation or deacetylation reactions. Decrotonylation/deacetylation was performed with 1.41 to 0.19 µM modified histone H3 and 0.03 µM HDAC1 (recombinant, human, Active Motif) at 30 °C and stopped at 95 °C for 1 min. Five different modified histone concentrations and five different time points were performed in triplicate for the decrotonylation and deacetylation reactions.

Each reaction was spotted in quadruplicate and left to dry before being rinsed with transfer buffer and TBS-T. Western blot was performed using anti-H3K18ac and anti-H3K18cr antibodies. Spots were quantified using image J and spot intensity was converted to substrate concentration by multiplying each relative value by the concentration of modified histone in the reaction mix. Substrate concentration against time were plotted for each histone concentration and a linear regression was fitted for the first 30 s to 1 min of the reaction, as appropriate. The rate of each reaction set was plotted as replicates against substrate concentration. GraphPad Prism Version 7 software was used to calculate $K_m$, $V_{max}$, and $K_{cat}$.

**Colorimetric deacetylation assay**. Synthesis of BOC-Lys(acetyl)-AMC and BOC-Lys(crotonyl)-AMC and the colorimetric assay was essentially as in ref. [2] with some modifications: for the synthesis of crotonyl-N-hydroxysuccinimide, N-hydroxysuccinimide (149 mg, 6 equivalents, eq) was dissolved in 2.5 ml anhydrous dichloromethane to which diisopropylethylamine (167 mg, 6 eq) was added. The solution was cooled in an ice bath and trans-crotonoyl chloride (135 mg, 6 eq) was added and the reaction stirred at 20 °C for 3 h. Ethyl acetate (5 ml) was added and the solution washed with 5 ml brine, dried over magnesium sulfate and concentrated. The crude product was dissolved in 3 ml dry ACN and the appropriate amount used in the next step. Acetyl-N-hydroxysuccinimide was prepared in the same way but using acetyl chloride rather than crotonoyl chloride. For the synthesis of BOC-Lys (crotonyl)-AMC, Boc-Lys-AMC acetate salt (100 mg, 1 eq) purchased from Bachem AG was dissolved in 4 ml water:ACN (1:1). Sodium bicarbonate (19.95 mg, 1.1 eq) was added in 1 ml water followed by trans-crotonyl-N-hydroxysuccinimide ester (2 eq) in 2 ml ACN. The reaction was left for 18 h and then concentrated to remove the ACN. The product was extracted into ethyl acetate, dried over anhydrous magnesium sulfate and concentrated. The product was purified on silica in dichloromethane:methanol (15:1). BOC-Lys(acetyl)-AMC

was prepared in the same way but using acetyl-*N*-hydroxysuccinimide rather than crotonyl-*N*-hydroxysuccinimide.

The enzymatic reaction was set up in a 96-well plate in triplicate in a 51-μl volume with 0.035 μM HDAC1 and 0.2 mM BOC-Lys(acetyl)-AMC/BOC-Lys (crotonyl)-AMC in HDAC buffer (see above) at 30 °C for 2 h. After this, 5 μl of a 1 mg/ml trypsin solution in 1 mM HCl was added and the mixture was incubated 1 h at 37 °C. Fluorescence was read with a Pherastar plate reader at 360 nm excitation/ 450 nm emission.

**Data availability**. Next-generation sequencing data have been submitted to GEO under accession code GSE96035. The authors declare that all other data supporting the findings of this study are within the manuscript and its supplementary files or are available from the corresponding authors upon request.

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

## Acknowledgements

We thank Drs. Simon Andrews, Anne Segonds-Pichon, and Felix Krueger, Babraham Bioinformatics, for help with data processing, statistical and bioinformatic analysis, Dr. Kristina Tabbada and Clare Murnane, Babraham Institute Next Generation Sequencing facility, for next-generation sequencing, Nicolas Le Novère for advice with the kinetic analysis, all the staff of Babraham Institute BSU for animal care, and Matthew Sale for HCT116 cells and help with the cell cycle experiments. P.V.-W. thanks Drs. Kazuyuki

Ohbo, Yokohama City University, and Masataka Nakamura, Tokyo Medical and Dental University, for hosting him for 2 months in 2012 in Japan, during which time the idea for this work was conceived. This work was funded by the UK Biotechnology and Biological Sciences Research Council (BBSRC), the UK Medical Research Council through project grant MR/N009398/1 to P.V.-W. and M.V., a Brazil-BBSRC Pump-priming award (BB/N013565/1) to P.V.-W. and M.A.R.V. and funding by the Science Policy Committee, Babraham Institute to P.V.-W. and M.V., Grants from NC3R (NC/L001217/1) to M.V. and FAPESP (2012/10653-9, 2015/50379-1 and 2015/14105-4) to M.A.R.V., grants from the Italian Association for Cancer Research (AIRC), the Italian Ministry of Health (RF-GR2011) and the EPIGEN flagship project grant to T.B. R.M., S.B., Z.H., M.S., C.M., and H.P. were funded through the Erasmus+ programme and H.B. through a BBSRC studentship.

## Author contributions

P.V.-W., T.B., M.A.R.V., J.D., and R.F. conceived the research and designed the experiments. A.C. and T.B. performed mass spectrometry analysis; P.J. performed ChIP-seq in colon epithelium; A.L. performed RNA-seq of colon epithelium; R.F., R.O.C., J.L.F., and M.A.R.V. performed antibiotics treatment experiments in mice and bacterial load measurements; W.R.R., F.T.S., and C.M.F. performed SCFAs measurements; J.K. performed immunohistochemistry; R.F., S.B., Z.H., and P.V.-W. performed in vitro enzymatic assays; C.S. and E.S. performed ChIP-seq and RNA-seq of HCT116 cells; E.S. performed the cell cycle experiments; J.C. performed chemical synthesis; R.F., J.D., C.S., E.S., A.L., H.B., J.K., R.O.C., J.L.F., F.T.S., H.P., M.S., R.M., C.M., and P.V.-W. performed further experiments; R.F., R.O.C., J.G., A.L., and M.V. set up gut organoid culture; R.F., C.S., P.J., A.L., C.M.F., A.C., T.B., M.A.R.V, and P.V.-W. performed the data analysis; P.V.-W. wrote the manuscript with inputs from almost all co-authors, especially R.F.

## Additional information

**Competing interests:** The authors declare that they have no competing financial interests.

