## [Peer Review File · Nature Communications]

Reviewers' Comments:

Reviewer #1:

Remarks to the Author:

Major:

1. The novelty is the major concern about this paper: (1) crotonylation on histones or non-histone proteins have been systematically identified in different cell lines by different groups (Minjia Tan et al., *Cell*, 2011; Wei Wei et al., *J. Proteome Res.*, 2017; Weizhi Xu et al., *Cell Res.*, 2017). Although the authors used three types of intestinal cells to profile crotonylation, they only identified 5 crotonylation sites, which is very preliminary and lacking careful analysis; (2) H3K18cr was also previously identified and characterized to affect gene transcription (Benjamin R. Sabari et al., *Mol. Cell.*, 2015). The authors profiled the genomic localization of H3K18cr again in intestinal cells, while they did not do any related experiment to show the functional roles of H3K18cr in this physiological context, thus provided not much information about this modification; (3) the authors showed that HDAC1-3 could remove crotonylation from histone proteins *in vitro*, while recently Weizhi Xu et al. already showed that HDAC1 and HDAC3 have de-crotonylation activities toward non-histone proteins in living cells.

2. The authors identified and confirmed H3K18cr as an abundant crotonylation site in intestinal cells, and showed that H3K18cr was associated with TSS and gene expression (Figure 1 and Supplementary Figure 5). This result seemed interesting, however the authors did not follow up on this modification, while they just roughly analyze the ChIP data and more focused on pan Kcr in next experiments. For example, in Figure 2e, it is more interesting to also examine the changing pattern of H3K18cr during cell cycle. As stated above, since H3K18cr is not a newly-identified and uncharacterized modification, it is more important and will provide more useful information if the authors perform more experiments to investigate the functional role of H3K18cr or Kcr in the intestine-specific biological events.

3. Some data is poorly organized in these paper. For example: (1) it is strange that anti-H4 blotting was used as loading control for anti-H3K18cr/H3K18ac (Figure 2a, Supplementary Figure 7 and 8), either direct staining of total histone protein or anti-H3 blotting is a better choice; (2) in Figure 3a, the author was trying to demonstrate that HDAC1-3 complex could remove crotonylation from histone proteins *in vitro*. In the left panel, HDAC1 and HDAC3 was examined against H3K18cr, pan Kcr and H3K18ac, while no loading control was performed. However, for HDAC2 (right panel), the authors used anti-H3 blotting as loading control, while only pan Kcr level was examined, no H3K18cr and H3K18ac (is it because HDAC2 showed no activity toward H3K18ac/cr in this experiment?). It should not be technically difficult to organize this result in one data piece for presence, so the authors should either explain it in text or re-organize these data; (3) considering the obvious difference in the Western blot result, it is not necessary to show the quantification result in the main text figure (Figure 2a-b, 2e and Figure 3b-c), especially when there was no error bar for it.

4. Another major concern is that the author only used inhibitor treatment to demonstrate the decrotonylation activity of HDACs (Figure 2b and Supplementary Figure 8). This is not convincing because of the off-target effect of inhibitors, especially at high concentration. Actually, in Supplementary Figure 8, the usage concentration of TSA and SAHA is mini-molar level, which is extremely higher than their IC₅₀ (nano-molar in cell free system; micro-molar in living cells). The authors claimed, without data shown, that knockdown of HDAC1-3 did not affect the global acetylation or crotonylation activity because of the redundancy between these enzymes and the impossibility of sufficient triple knockdown of HDAC1-3 (Line 127-130). In fact, the knockdown data is crucial for the evaluation of HDAC1-3 as regulation enzymes of histone crotonylation in living cells, the authors should include these data (positive or negative) in the manuscript. Actually, except for overall Kcr level, the authors could also examine the site-specific crotonylation level upon HDAC1-3 knockdown. In addition, overexpression of HDACs in cells is another choice to investigate the HDACs decrotonylation activities in cellular context.

5. In Figure 3b-c, the authors investigated the *in vitro* activities of HDAC1-3 toward histone crotonylation and acetylation, however they did anti-H3K18ac and anti-Kcr blotting parallelly. If

they want demonstrated the “similar kinetic” (Line 135) or the “differential impact of inhibitors on HDAC” (Line 139) of HDACs decrotonylation and deacetylation activities, they at least should perform the anti-H3K18ac and anti-H3K18cr (or anti-Kac and anti-Kcr) blotting parallelly, otherwise it is not comparable.

Minor:

1. The pan Kcr blotting showed inconsistency in different figure, the author should explain this. For example, in Figure 2b, 2e and Supplementary Figure 4b, 7a, the anti-Kcr blotting showed H3 and H4 two bands, while only showed one band in Figure 3.
2. The data “not shown” situation (Line 130 and 137) should always be avoided, every statement in the text should be supported by appropriate experimental data.
3. Line 41: “Histone post-transcriptional modification” probably should be “post-translational”.
4. Figure 2e (Top panel): It should be indicated that how the cell percentage in different cell cycle phase was determined. This should be mentioned in method session, and protein markers for S phase (e.g. cyclin B) or G2/M (e.g. H3S10ph) phase should be included in Western blot analysis.
5. Line 124-125: “This suggests that histone crotonylation may be linked to nucleosome assembly, as has been shown for H3K18ac.....” There was no data supporting this statement. The data from Figure 2e only indicated potential roles of H3K18cr during cell cycle, while is far away from drawing the conclusion of “nucleosome assembly”.
6. The data in Supplementary Figure cannot confirm the specificity of the anti-Kcr and anti-H3K18ac antibodies. There is no evidence to show the site-specificity and modification-specificity of those antibodies, e.g. whether the pan anti-Kcr antibody can differentiate Kbu from Kcr?
7. There is no kinetics (i.e. Km, Kcat/Km) analysis of HDAC1-3 towards Kcr histone proteins/peptide.
8. The author demonstrates that “the HDACs require the peptide context of histone H3 form efficient decrotonylation activity”. To support this idea, the author can synthesize several peptides with different context around Kcr to test the sequence dependence of HDACs decrotonylation activity.
9. There is no relative abundance percentage data of peptides containing only H3K8cr. What is the relative abundance of H3K8cr compared with H3K18crK23ac?

Reviewer #2:

Remarks to the Author:

Acetylation-like acetylations, such as crotonylation, have been identified in the past few years, but one major issue to be addressed is the functional relevance. In this paper, the authors have characterized histone H3 crotonylation at lysine 18 in the small intestine crypt and colon, as well as in cultured colonic cells. The results are very convincing and provide important information about the potential biological function. Once published, the results should be valuable to researchers in this and related fields. I would recommend the publication of the manuscript in Nat. Commun., pending the following modifications.

- 1) It is important to analyze H3K18 crotonylation in one or two other tissues as control to show whether the abundant H3K18cr is specific to the small intestine crypt and colon.
- 2) The manuscript is well written and contains 3 main figures and 10 supplemental figures. To improve the readability, I would suggest to move some important data to the main figure section. For example, Figs S1 and S5 can be reorganized with Fig. 1 to form two new main figures. Fig. S9 can also be moved to the main figure section. This way, readers will not need to go to the supplemental figure section (which tends to be ignored frequently) to get the key points of the paper.
- 3) The cell cycle angle is weak but it is highlighted in the title. I would suggest to change the title to: “Histone H3K18 decrotonylation in the small intestine crypt and colon by class I histone

deacetylases." This will also highlight the main points of the paper.

4) In the abstract, replace "HDAC1, 2 and 3" with "HDAC1, HDAC2 and HDAC3" to increase the visibility when readers search through PubMed using the keywords HDAC1, HDAC2 and HDAC3.

5) Ensure to include the GEO accession number on page 41 (middle).

Reviewer #3:

Remarks to the Author:

Review of NCOMMS-17-06573: "Pervasive histone decrotonylation is cell cycle regulated by class I histone deacetylases" by Denizot et al.

Summary:

By performing mass spec on histone H3 from mouse small intestine, colon and crypts the authors identified histone crotonylation of H3K18 (Fig 1). The authors perform ChIP-seq in colon epithelial cells for H3K18cr and determine its association with active promoters which is consistent with previous reports (Fig 1e, f). In Figure 2 the authors show that inhibition of HDACs leads to increases in histone acylation of H3K18 (both butyl and crotonyl) and not just acetylation. In Figure 3 the authors show that several class I HDACs can function as histone decrotonylases.

General comments:

The core finding of this manuscript, that histone crotonylation of H3K18 can be removed by several class I HDACs, is highly interesting finding that would be of general interest to the chromatin field. However, there are a variety of issues with the manuscript that indicate that publication in its current form would be premature.

1) The manuscript is written without much depth or context. There is very little in the way of an introduction setting the stage and very little in the way of discussion placing the results in the context of the field.

2) The core conclusion of the manuscript, that class I HDACs mediate decrotonylation, is not characterized as rigorously as a such a potentially groundbreaking claim should warrant (e.g. the deacetylation and decrotonylation activities are only qualitatively compared when a more rigorous and quantitative kinetic analysis can be performed with the techniques used in the manuscript.)

3) The thesis statement as expressed in the title is not fully supported by the data presented in the manuscript. While the authors show in one panel (Fig 2e) that H3K18cr levels vary in a cell cycle-dependent fashion, it is completely unclear that this is due to regulation of HDACs, other enzymes that might regulate decrotonylation or the enzymes (perhaps p300) that establish the mark.

Specific comments:

4) The mass spec data across the three gut compartments is largely negative data. There are no statistical differences in crotonylation demonstrated.

5) The H3K18cr ChIP-seq data largely confirms what has been seen previously in other cell types and does not add to the conclusions of the paper.

6) On page 3 the authors in referring to Fig 1F state that H3K18cr coincides with only a subfraction of H3K4me3-marked promoters. It is unclear if the authors are suggesting that these different subfractions represent distinct gene regulatory states of chromatin or (more likely) that the sensitivity of the H3K18cr ChIP-seq data is lower than the H3K4me3 data. If the former, the

graphical representation is insufficient to differentiate between those two possibilities. Additionally, it is not clear how either possibility substantively adds to the manuscript, without additional experiments.

7) In Figure 2 the authors supplement cells with crotonic acid as a way to upregulate the level of crotonyl-coA, as previously reported, and demonstrate an increase in cronylation of H3K18, However, given the effect of class I HDAC inhibitors, including the chemically similar butyric acid, the authors should test if crotonic acid is simply an HDAC inhibitor as well.

8) Figure 3, which should be the jewel of the manuscript, is not as rigorous as it should be. The data seem to indicate that HDACs are more active on H3K18cr than H3K18ac, which is a striking finding. However, this conclusion requires more rigorous enzyme kinetics. Km and other kinetic parameters should be calculated. Ki or IC50 for the inhibitors on deacetylation and decrotonylation should be calculated. Given that TSA, and most HDAC inhibitors, are active site binders which prevent substrate binding, reasonable explanation for the apparent specificity of TSA for maintaining H3K18ac vs H3K18cr would be that H3K18cr has a relative higher affinity for the enzyme. Determining the enzyme kinetics using both substrates could confirm this.

9) I suggest figure S10, which invalidates a previous assay used to discount class I HDACs as decrotonylases, be brought to the main figures, as explaining this discrepancy with previously published work is critical to the conclusions of the manuscript.

We are grateful for the constructive suggestions of all reviewers. We have addressed all comments and especially we have implemented reviewer 1's suggestion to strengthen the link between histone crotonylation and intestinal biology:

Using antibiotics treatment of mice, we now provide evidence that histone crotonylation is strongly and globally affected by the presence of microbiota. Our work suggests that the microbiota affect histone crotonylation, at least in part, through the inhibitory effect of SCFAs on class I HDACs.

We feel these new data significantly strengthen our manuscript and are of great interest to a broad readership, linking histone modifications to the microbiota. Furthermore, these findings extend our work beyond the simple analysis of interaction between class I HDACs and crotonylation. In summary, we believe our study will be of broad interest and high impact. Below are point-by-point responses to the reviewers' comments.

The paper is now Fellows et al. instead of Denizot et al., as Rachel Fellow now provided the majority of contributions. All authors agreed to the change in author list and contributions.

Reviewer1

Reviewer 1, major comments 1 and 2: This reviewer suggests strengthening the manuscript by investigating “the functional role of H3K18cr or Kcr in the intestine-specific biological events”. Furthermore, the reviewer suggests to “examine the changing pattern of H3K18cr during cell cycle”.

Our response: We have strengthened the link between histone crotonylation and intestinal biology by examining how depleting the microbiota in the mice using antibiotics affects histone crotonylation. We find that this treatment leads to a significant, global loss of histone H3K18 and H4 crotonylation

(H4K8cr) and is linked to a surprising upregulation of HDAC2 levels in the colon (new Figure 3). These data link the class I HDACS to histone crotonylation in a biological context. Furthermore, our data suggest that the microbiota affect histone crotonylation, most likely through their generation of short chain fatty acids. Abstract, introduction, results and discussion sections have been modified to reflect these new findings. Our new data that show that histone crotonylation is especially abundant in the colon and brain which also strengthens the biological link (new Figure 1a, see response to comment 1, reviewer 2).

As suggested, we also examined the changing pattern of H3K18cr during cell cycle, new Figure 6. These new data strengthen the link between histone crotonylation, including H3K18cr, during the cell cycle and class I HDACs.

We write:

'We performed cell cycle arrest and release experiments in presence or absence of class I HDAC specific inhibitor MS275. The results indicated that histone crotonylation, including H3K18cr are linked to cell cycle progression, showing an increase in S and G2-M phase over G1 arrested cells (Figure 6, lanes 1-8). This experiment also suggests that class I HDACs may be involved in this cell cycle mediated modulation of histone crotonylation levels, as the G1-arrest linked downregulation of histone crotonylation is inhibited in the presence of MS275 compared to untreated cells (Figure 6, lanes 1,2,9,10). In addition, in MS275-treated compared to untreated cells histone crotonylation was modulated during the cell cycle, but levels remained generally higher (Figure 6, lanes 9-16).'

While, as reviewer 1 points out, several studies have made the connection between class I HDACs and crotonylation, our study brings this into an exciting biological context, the intestinal epithelium and its link to the microbiota. Furthermore, additional data, including new data in the revised manuscript, complement and extend other studies in important and exciting ways:

(1) We link for the first time histone crotonylation to the cell cycle (Figure 6); (2) we provide the first study of the effect of HDAC inhibition on the distribution of H3K18cr versus H3K18ac genome-wide (Figure 5); (3) we provide the first enzymatic, kinetic data on decrotonylation versus deacetylation (Figure 7b); (4) we show that crotonic acid not only inhibits deacetylase and decrotonylase activity of HDAC1, but at high concentration of crotonic acid, HDAC1 mediates histone crotonylation (Figure 7d); (5) importantly, our *in vitro* data using purified enzymes show that the microbiota generated SCFA butyrate directly inhibits decrotonylation by class I HDACs, linking histone crotonylation to microbiota (Figure 7c).

Reviewer 1, list of issues under major comment 3:

(1) The reviewer writes: 'it is strange that anti-H4 blotting was used as loading control for anti-H3K18cr/H3K18ac (Figure 2a, Supplementary Figure 7 and 8), either direct staining of total histone protein or anti-H3 blotting is a better choice'.

Our response: We now show an improved figure corresponding to what was Figure 2a (now Figure 4a) with histone H3 as loading control. However, histone H4 is an obligatory partner of histone H3 and is thus a valid loading control. Therefore, we did not change the supplementary figures.

(2) 'In Figure 3a, the author was trying to demonstrate that HDAC1-3 complex could remove crotonylation from histone proteins in vitro. In the left panel, HDAC1 and HDAC3 were examined against H3K18cr, pan Kcr and H3K18ac, while no loading control was performed. However, for HDAC2 (right panel), the authors used anti-H3 blotting as loading control, while only pan Kcr level was examined, no H3K18cr and H3K18ac (is it because HDAC2 showed no activity toward H3K18ac/cr in this experiment?). It should not be technically difficult to organize this result in one data piece for presence, so the authors should either explain it in text or re-organize these data;'

Our response: We have followed this reviewer's advice, repeated the experiment and present the data as suggested in new Figure 7a.

(3) 'Considering the obvious difference in the western blot result, it is not necessary to show the quantification result in the main text figure (Figure 2a-b, 2e and Figure 3b-c), especially when there was no error bar for it.'

Our response: We removed these quantifications from the figures.

Reviewer 1, major comment 4: This reviewer is concerned about off-target effects of using HDAC inhibitors in our assays and suggests showing RNAi knockdown or HDAC over-expression experiments.

Our response: Actually, we used TSA and SAHA in submicromolar amounts in the cell assays and have clarified the labeling on this figure (now supplementary Figure 9). The sub-micromolar concentrations of MS275, TSA and SAHA that we use very strongly implicate class I HDACs in de-crotonylation.

We have performed HDAC1 over-expression experiments as suggested and these show the over-expression of HDAC1, but not a mutant form or a GFP control affects histone crotonylation in cells in culture (new Supplementary Fig. 10). As we show the over-expression data, we have removed the negative RNAi experiments.

Reviewer 1, major comment 5: 'In Figure 3b-c, the authors investigated the in vitro activities of HDAC1-3 toward histone crotonylation and acetylation, however they did anti-H3K18ac and anti-Kcr blotting parallelly. If they want to demonstrate the "similar kinetic" (Line 135) or the "differential impact of inhibitors on HDAC" (Line 139) of HDACs decrotonylation and deacetylation activities, they at least should perform the anti-H3K18ac and anti-H3K18cr (or anti-Kac and anti-Kcr) blotting parallelly, otherwise it is not comparable.'

Our response: We have now performed a much more thorough kinetic analysis of decrotonylation versus deacetylation of H3K18 (new Figure 7b), providing estimates for Vmax, Km and Kcat values.

We write:

'Using this *in vitro* approach, we estimated enzyme kinetic values for the decrotonylation of H3K18cr and deacetylation of H3K18ac and found that HDAC1 has a similar capacity to perform each reaction. HDAC1 has lower Vmax and Kcat values for the decrotonylation compared to the deacetylation reactions indicating that HDAC1 has a higher maximum turnover with the acetyl substrate than the crotonyl substrate (Fig. 7b, Supplementary Fig. 11). We found that the Km with acetylated histones of 1.47 μM was similar to a published Km of HDAC with a BOC-lys(acetyl)-AMC of 3.7 μM ²⁰. Published Km's with different fluorescent or tritiated acetyl substrates vary from 0.68 μM to 78 μM ^{21,22,23,24}. Relative to this, the Km of the crotonyl substrate was lower at 0.42 μM suggesting that HDAC1 can efficiently bind crotonylated histones at low concentrations of substrate. This may be relevant in the context of the cell where HDAC1 could respond to minor fluctuations in the availability of the crotonyl-substrate.'

Minor:

Reviewer 1, minor comment 1. 'The pan Kcr blotting showed inconsistency in different figures, the author should explain this. For example, in Figure 2b, 2e and Supplementary Figure 4b, 7a, the anti-Kcr blotting showed H3 and H4 two bands, while only showed one band in Figure 3.'

Our response: In what was Figure 3, we crotonylated recombinant histone H3, there is no histone H4 in the reaction. Thus, no H4 band would be expected.

Reviewer 1, minor comment 3. 'Line 41: "Histone post-transcriptional modification" probably should be "post-translational".'

Our response: This has now been corrected.

Reviewer 1, minor comment 4: 'Figure 2e (Top panel): It should be indicated that how the cell percentage in different cell cycle phase was determined. This should be mentioned in method session, and protein markers for S phase (e.g. cyclin B) or G2/M (e.g. H3S10ph) phase should be included in Western blot analysis.'

Our response: We used propidium iodide staining to determine DNA content using flow cytometry and determined cell cycle profiles using the Cell Cycle Tool in the FlowJo software. This is highly established and accepted and the use of additional cell cycle markers is not necessary. We apologize that we did not elaborate this in the method section, which we have now corrected.

Reviewer 1, minor comment 5: 'Line 124-125: "This suggests that histone crotonylation may be linked to nucleosome assembly, as has been shown for H3K18ac....." There was no data supporting this statement. The data from Figure 2e only indicated potential roles of H3K18cr during cell cycle, while it is far away from drawing the conclusion of "nucleosome assembly.'

Our response: We removed the statement about nucleosome assembly.

Reviewer 1, minor comment 6: ‘The data in Supplementary Figure cannot confirm the specificity of the anti-Kcr and anti-H3K18ac antibodies. There is no evidence to show the site-specificity and modification-specificity of those antibodies, e.g. whether the pan anti-Kcr antibody can differentiate Kbu from Kcr?’

Our response: For anti-H3K18ac we use an antibody from ABCAM (ab1191) where the site specificity has been well validated by the company using peptide competition assays. This antibody has been widely used with >60 references citing it. The anti-Kcr antibody has been characterized by the Zhao group (Tan et al., Cell 2011) which shows that it does not bind peptides with butyrylated lysines (Figure 4a of this paper).

Reviewer 1, minor comment 7: ‘There is no kinetics (i.e. Km, Kcat/Km) analysis of HDAC1-3 towards Kcr histone proteins/peptide.’

Our response: We have performed this now with HDAC1 (see above), new Figure 7b. We also show HDAC1-3’s activity in decrotonylation/deacetylation using antibodies against H3K18cr vs H3K18ac (new Figure 7a). See response to major comment 5.

Reviewer 1, minor comment 8: ‘The author demonstrates that “the HDACs require the peptide context of histone H3 form efficient decrotonylation activity”. To support this idea, the author can synthesize several peptides with different context around Kcr to test the sequence dependence of HDACs decrotonylation activity.’

Our response: In the paper we only wished to suggest that peptide context specificity may explain why the decrotonylase activity was not shown with the synthetic substrates (in contrast to the deacetylase activity). We appreciate this reviewer’s experimental suggestion, but suggest that this should be followed up in future studies. We now write:” Future studies will examine if HDACs require the peptide context of histone H3 for efficient decrotonylation activity.”

Reviewer 1, minor comment 9: ‘There is no relative abundance percentage data of peptides containing only H3K8cr. What is the relative abundance of H3K8cr compared with H3K18crK23ac?’

Our response: These data are in Supplementary table 1. This shows that H3K18crK23ac is significantly more abundant than H3K18cr alone (e.g., in the colon fraction, H3K18crK23ac: ~ 2%, H3K18cr alone: ~0.2%).

Reviewer 2

Reviewer 2 comment 1: ‘It is important to analyze H3K18 crotonylation in one or two other tissues as control to show whether the abundant H3K18cr is specific to the small intestine crypt and colon.’

Our response: We have performed an analysis of the level of histone crotonylation in several tissues (colon, brain, liver, spleen, kidney) using the antibodies against crotonyl-lysine and against H3K18cr and western blot of whole tissue extracts. This indicates greatest levels of histone crotonylation in brain and colon among the tissues analyzed. The data are shown in new Figure 1a. This also revealed the presence of an approximately 70 kDa protein in the brain extract that is recognized by the antibody against crotonyl-lysine, indicating the presence of a crotonylated nonhistone protein in brain.

Reviewer 2 comment 2: 'To improve the readability, I would suggest to move some important data to the main figure section. For example, Figs S1 and S5 can be reorganized with Fig. 1 to form two new main figures. Fig. S9 can also be moved to the main figure section. This way, readers will not need to go to the supplemental figure section (which tends to be ignored frequently) to get the key points of the paper.'

Our response: We followed this advice.

Reviewer 2 comment 3: 'The cell cycle angle is weak but it is highlighted in the title. I would suggest to change the title to: "Histone H3K18 decrotonylation in the small intestine crypt and colon by class I histone deacetylases." This will also highlight the main points of the paper.'

Our response: We changed the title to:

Histone decrotonylation in the intestinal epithelium by class I histone deacetylases

Reviewer 2 comment 4: 'In the abstract, replace "HDAC1, 2 and 3" with "HDAC1, HDAC2 and HDAC3" to increase the visibility when readers search through PubMed using the keywords HDAC1, HDAC2 and HDAC3.'

Our response: We followed this advice.

Reviewer 2 comment 5: 'Ensure to include the GEO accession number on page 41 (middle).'

Our response: We followed this advice.

Reviewer 3

Reviewer 3 comment 1: 'The manuscript is written without much depth or context. There is very little in the way of an introduction setting the stage and very little in the way of discussion placing the results in the context of the field.'

Our response: We have now improved the introduction and discussion. The field of alternative histone acylations is rapidly developing and more papers highlighting the biological potential of these modifications have been added since we submitted the first manuscript. We now refer to many of these papers.

Reviewer 3 comment 2: ‘The core conclusion of the manuscript, that class I HDACs mediate decrotonylation, is not characterized as rigorously as a such a potentially groundbreaking claim should warrant (e.g. the deacetylation and decrotonylation activities are only qualitatively compared when a more rigorous and quantitative kinetic analysis can be performed with the techniques used in the manuscript.)’

Our response: We now include a better characterization, including kinetic analysis in the paper, new Figure 7b; please see our response to major comment 5 of reviewer 1.

Reviewer 3 comment 3: ‘The thesis statement as expressed in the title is not fully supported by the data presented in the manuscript. While the authors show in one panel (Fig 2e) that H3K18cr levels vary in a cell cycle-dependent fashion, it is completely unclear that this is due to regulation of HDACs, other enzymes that might regulate decrotonylation or the enzymes (perhaps p300) that establish the mark.’

Our response: We removed the reference to cell cycle in the title, but also include a new Figure 6 that strengthens the link between cell cycle regulation of histone crotonylation and class I HDAC activity (see also response to reviewer 1).

Reviewer 3, specific comment 4: ‘The mass spec data across the three gut compartments is largely negative data. There are no statistical differences in crotonylation demonstrated.’

Our response: We agree with the referee, but we also do not make much out of the differences that we observe.

Reviewer 3, specific comment 5: ‘The H3K18cr ChIP-seq data largely confirms what has been seen previously in other cell types and does not add to the conclusions of the paper.’

Our response: One of the important findings of our study is that H3K18 crotonylation is a surprisingly abundant modification in the intestinal epithelium, not a trace modification, and, thus, a characterization of this modification by ChIPseq in the colon epithelium is warranted. In fact, reviewer 1 writes: ‘The authors identified and confirmed H3K18cr as an abundant crotonylation site in intestinal cells, and showed that H3K18cr was associated with TSS and gene expression (Figure 1 and Supplementary Figure 5). This result seemed interesting’. Thus, we wish to show these data.

Reviewer 3, specific comment 6: ‘On page 3 the authors in referring to Fig 1F state that H3K18cr coincides with only a subfraction of H3K4me3-marked promoters. It is unclear if the authors are suggesting that these different subfractions represent distinct gene regulatory states of chromatin or (more likely) that the sensitivity of the H3K18cr ChIP-seq data is lower than the H3K4me3 data. If the former, the graphical representation is insufficient to

differentiate between those two possibilities. Additionally, it is not clear how either possibility substantively adds to the manuscript, without additional experiments.'

Our response: We agree with the referee and now write simply: 'This ChIP-Seq analysis showed that H3K18cr is associated with transcription start sites (TSS) (Figure 2a-d), similar to H3K4me3 (Figure 2c), as has been shown before in macrophages⁶.'

Reviewer 3, specific comment 7: 'In Figure 2 the authors supplement cells with crotonic acid as a way to upregulate the level of crotonyl-coA, as previously reported, and demonstrate an increase in crotonylation of H3K18. However, given the effect of class I HDAC inhibitors, including the chemically similar butyric acid, the authors should test if crotonic acid is simply an HDAC inhibitor as well.'

Our response: We now show that crotonic acid not only inhibits deacetylase and decrotonylase activity of HDAC1 *in vitro*, but at high concentration of crotonic acid, HDAC1 mediates histone crotonylation (new Figure 7c.d). We write:

'Remarkably, at crotonate concentrations of 6 mM or higher, HDAC1 catalyzed the addition of the crotonyl moiety to lysine groups of histones (Figure 7d). This did not occur with acetate or butyrate, possibly due to the higher chemical reactivity of crotonate. While this finding may not be directly biologically relevant, as it is unlikely that such high concentrations of crotonate are found intracellularly, it highlights the reversible nature of the decrotonylation reaction by HDAC1.'

Reviewer 3, specific comment 8: 'Figure 3, which should be the jewel of the manuscript, is not as rigorous as it should be. The data seem to indicate that HDACs are more active on H3K18cr than H3K18ac, which is a striking finding. However, this conclusion requires more rigorous enzyme kinetics. Km and other kinetic parameters should be calculated. Ki or IC50 for the inhibitors on deacetylation and decrotonylation should be calculated. Given that TSA, and most HDAC inhibitors, are active site binders which prevent substrate binding, reasonable explanation for the apparent specificity of TSA for maintaining H3K18ac vs H3K18cr would be that H3K18cr has a relative higher affinity for the enzyme. Determining the enzyme kinetics using both substrates could confirm this.'

Our response: We now include a better characterization, including kinetic analysis of decrotonylation versus deacetylation, new Figure 7b,c. We provide estimates for Vmax, Km, Kcat and IC50s as requested (See also response to Reviewer 1 (major comment 5, minor comment 7)).

Reviewer 3, specific comment 9: 'I suggest figure S10, which invalidates a previous assay used to discount class I HDACs as decrotonylases, be brought to the main figures, as explaining this discrepancy with previously published work is critical to the conclusions of the manuscript.'

Our response: We followed this advice, this figure is now main Figure 8.

Reviewers' Comments:

Reviewer #1:

Remarks to the Author:

The author characterized histone crotonylation at histone H3 lysine 18 in intestinal epithelia and found that this highly dynamic histone mark under the regulation of HDACs, which, so far, are known as lysine deacetylase. To strengthen the link between histone crotonylation and intestinal biology, the author performed the antibiotics treatment of mice, which showed the effect of microbial environment on the level of histone lysine crotonylation. In addition, the author provide evidence to show that inhibitory effects of SCAFs on HDACs enzymatic activity. However, there are still several major issues needed to address before its publication.

Major Comments:

1. In revised paper, the authors linked the intestinal microbiota and the regulation of crotonylation together. (1) The authors showed that the depletion of gut microbiota led to the decrease of H3K18cr and H4K8cr, however the biological significance of H3K418cr or H4K8cr was still not be illustrated. For example, whether the crotonylation is essential for the physiological functions of intestine, or whether the microbiota-depletion-induced decrease of crotonylation is related to the inflammation or diseases caused by the dysregulation of microbiota. Given the fact that the author is capable of performing ChIP-seq using colon epithelium tissues (Fig 2), it would be helpful to perform ChIP-seq using the antibiotic-treated tissues to further elucidate the functional roles of H3K18cr in the intestinal-related physiological events.
2. In in Fig 5, the authors showed that HDAC inhibitor MS275-treatment reduced the TSS-enrichment of H3K18cr genome-widely. It is not convincing that the authors reasoned that it is because the "overall increase of these marks over the genome outside of the TSS regions", because the H3K18cr level on TSS should also increase upon MS275. More explanations or experiment should be included to this issue.
3. The cell cycle part (Fig 6) is somewhat incongruous in this story, and the data should be interpreted more seriously.
4. The author claimed that H3K18cr, in association with H3K23 acetylation is the most abundant histone crotonylation mark in crypt and colon fractions. The percentage of peptide with only K18cr but not K23ac, even a little higher than other Kcr peptides, but much lower than H3K18CrK23Ac. Since HDACs have been identified as lysine deacetylase, it is highly possible HDAC enzymes can modulate the dynamics of H3K23ac upon the change of intestinal microbial environment. There was no data to demonstrate the effects on H3K23ac in all experiments. Whether HDACs prefer only H3K18cr to H3K18crK23 combination or not, and whether H3K23ac plays a positive or negative role in modulating HDACs enzymatic activity toward H3K18cr? Those questions need to be addressed.
5. The author demonstrated the association between histone crotonylation and intestinal biology via regulating HDACs enzymatic activity, and showed the enrichment of H3K18cr on cancer-related genes. However, what is the effect of change of intestinal microbial environment on genomic distribution of H3K18cr mark and HDACs? Does it affect the expression level of cancer-associated genes? The author can perform ChIP-seq experiments using anti-H3K18cr and anti-HDACs antibodies to address those issues.

Minor Comments:

1. Since the anti-H3K18ac and anti-H3K18cr antibodies bought from companies, there is no need to give unnecessary details of those antibodies. But commercial information of antibodies used in this work should be provided in "Material and Methods" section.
2. In page 54, line 887, it should be lysine acetylation instead of monomethylation.
3. In page 62, line 1064, it should be H3K4me3 instead of H34me3.
4. In the "Material and Methods" section, mass data analysis of histone crotonylation was missing.
5. In page 54, " MS data analysis and relative abundance profiling" section, the explanation on D3-acetylation was missing.

Reviewer #2:

Remarks to the Author:

The authors have addressed my concerns and I thus recommend publication of the manuscript.

Reviewer #3:

Remarks to the Author:

This reviewer is satisfied with the author's responses to my critiques. The Wei et al paper (ref 28) does deal a blow to the novelty of some of the findings. However, even considering that paper this manuscript still presents several novel concepts. Most notably, the role of the microbiome in regulating Kcr, the differential effect of HDACi on Kcr vs Kac and the differential enzyme kinetics of HDAC towards Kcr vs Kac substrates.

Reviewer #4:

Remarks to the Author:

Fellow et al.

Histone decrotonylation in the intestinal epithelium by class I histone deacetylases

Reversible histone crotonylation is a chromatin modification linked to the cellular metabolism and has potential positive impact on transcription. Its biological and physiological role is largely unexplored. In an attempt to study tissue-specific effects of histone crotonylation the authors identified intestine and brain as tissues with particularly high levels of histone crotonylation. They show by ChIP-seq co-occupancy of regulatory regions by H3K18cr and H3K168ac. Based on HDAC inhibitor sensitivities and tests with purified enzymes they identify class I deacetylases HDAC1, HDAC2 and HDAC3 as relevant decrotonylases. Interestingly, short fatty acids such as butyrate, which are known natural HDAC inhibitors induce histone crotonylation. Based on the fact that butyrate and other short fatty acids are produced by microbes in the gut and intestine the authors show that depletion/reduction of the gut microbiome results in decreased histone crotonylation indicating a regulatory function of microbiome-produced short fatty acids on histone crotonylation and potentially on gene expression in intestinal cells.

Comments:

Given that this manuscript was previously reviewed by three experts I have only a major comment and few minor points.

The mouse experiments (depletion of the gut microbiome) significantly add to the importance of the manuscript because they suggest a biological significance of histone crotonylation with respect to intestinal homeostasis and maybe also to cancer.

Minor points:

Figure 3: In addition to the inhibitory function on class I HDACs short fatty acids produced by gut microbes also regulate the protein level of HDACs. HDAC2 levels are lower in the presence of an intact microbiome and elevated upon treatment with antibiotics. Kraemer et al. (2003) have shown that the carboxylic acids valproic acid and butyrate reduce the stability of the HDAC2 enzyme. This should be mentioned.

Figure 6: The cell cycle regulation of histone crotonylation is interesting. Class I HDACs have been implicated in the deacetylation of de novo synthesized HAT B-acetylated core histones (for instance: Bashkara et al. 2010). The MS-275 experiments indicate that reduced histone crotonylation is not only due to dilution of existing histones by non-crotonylated de novo synthesized histones but by an active process mediated by class I HDACs. This is an interesting finding which could be discussed in the manuscript.

Rebuttal letter to revision:

Microbiota derived short chain fatty acids promote histone crotonylation in the colon through histone deacetylases

Fellows et al., NCOMMS-17-06573B

~~We are again grateful to all reviewers for their constructive feedback and suggestions, which all led to an improved paper. Below we address these remaining issues.~~

Reviewer Nr. 1

Major Comment 1: In revised paper, the authors linked the intestinal microbiota and the regulation of crotonylation together. (1) The authors showed that the depletion of gut microbiota led to the decrease of H3K18cr and H4K8cr, however the biological significance of H3K418cr or H4K8cr was still not be illustrated. For example, whether the crotonylation is essential for the physiological functions of intestine, or whether the microbiota-depletion-induced decrease of crotonylation is related to the inflammation or diseases caused by the dysregulation of microbiota. Given the fact that the author is capable of performing ChIP-seq using colon epithelium tissues (Fig 2), it would be helpful to perform ChIP-seq using the antibiotic-treated tissues to further elucidate the functional roles of H3K18cr in the intestinal-related physiological events.

Our response:

We appreciate this reviewer's interest and eagerness to further understand the roles of histone crotonylation in the intestinal epithelium. We provide insights into this by combining our ChIP-seq analysis of H3K18cr with RNA-Seq data in colon epithelium, providing a first evidence of functional relevance of this modification in the intestine. However, the main focus of our current study is the regulation of histone crotonylation by short chain fatty acids and HDACs in the intestinal epithelium. The analysis of histone crotonylation in inflammation and disease will require much and careful work and will be the focus of future studies from our laboratories. In response to the thoughts of this reviewer, we write in DISCUSSION (page 13, lines 307-309): "Future work will explore further the roles of histone crotonylation in normal gut physiology including host-microbiome interaction, inflammation and disease".

Major Comment 2: In Fig 5, the authors showed that HDAC inhibitor MS275-treatment reduced the TSS-enrichment of H3K18cr genome-widely. It is not convincing that the authors reasoned that it is because the "overall increase of these marks over the genome outside of the TSS regions", because the H3K18cr level on TSS should also increase upon MS275. More explanations or experiment should be included to this issue.

Our response: To clarify, we write now (page 7, 168-177): "To investigate how HDAC inhibitors change the distribution of histone crotonylation over the genome, we performed ChIP-Seq analysis of H3K18cr and H3K18ac which showed a relative decrease of read counts over TSS when HCT116 cells were treated with MS275. This could be explained with an overall increase of these marks over the genome outside of the TSS regions (Figure 5a, b). ChIP-seq measures the relative proportional distribution of a mark over the genome. If the proportional increase of H3K18cr over TSS is lower than the one outside of TSS, an increase of histone crotonylation over regions outside of transcription start sites, TSS (which cover a much greater

proportion of the genome than all TSS combined) will lead to an apparent proportional drop of this mark in TSS region.”

3. The cell cycle part (Fig 6) is somewhat incongruous in this story, and the data should be interpreted more seriously.

Our response: We have now improved the discussion of this aspect of our work, also in response to the comments of reviewer number 4. We write (page 12 onwards, lines 284-299):

“Newly synthesized histones are acetylated on several, specific lysine residues (e.g., K5 and K12 on histone H4; K14, K18 on histone H3) prior to their deposition onto nascent chromatin^{26,27}, reviewed in²⁸. These acetylations are then globally removed following nucleosome assembly, which is important for the maintenance of repressive chromatin, such as pericentromeric heterochromatin^{19,29}. Class I HDACs are targeted to replicating chromatin and mediate this deacetylation^{30,31}. Therefore, the levels of these histone acetylations are modulated in a cell cycle and HDAC-dependent manner. In this study, we provide evidence that histone crotonylation marks behave similar to the pre-deposition acetylation marks: They are low in G1 arrested cells and increase as cells progress through S phase and this modulation depends on class I HDACs. Therefore, histone crotonylation is not simply modulated through dilution with non-crotonylated histones through the cell cycle, but appears to be actively regulated by HDACs. Future studies will address if histone crotonylation is also mediated prior to their deposition onto nascent chromatin and will address how HDACs and other factors regulate histone crotonylation concomitant with chromatin replication”.

Major Comment 4: The author claimed that H3K18cr, in association with H3K23 acetylation is the most abundant histone crotonylation mark in crypt and colon fractions. The percentage of peptide with only K18cr but not K23ac, even a little higher than other Kcr peptides, but much lower than H3K18CrK23Ac. Since HDACs have been identified as lysine deacetylase, it is highly possible HDAC enzymes can modulate the dynamics of H3K23ac upon the change of intestinal microbial environment. There was no data to demonstrate the effects on H3K23ac in all experiments. Whether HDACs prefer only H3K18cr to H3K18crK23 combination or not, and whether H3K23ac plays a positive or negative role in modulating HDACs enzymatic activity toward H3K18cr? Those questions need to be addressed.

Our response:

The referee raises a series of interesting questions regarding the physiological relevance of H3K18crH3K23ac. However, as already stated in the response to the Major Comment 1, this kind of analysis will require a much deeper consideration and is out of the focus of the present study where we report the role of HDACs as histone decrotonylases and the link of this histone modification to the intestinal microenvironment.

5. The author demonstrated the association between histone crotonylation and intestinal biology via regulating HDACs enzymatic activity, and showed the enrichment of H3K18cr on cancer-related genes. However, what is the effect of change of intestinal microbial environment on genomic distribution of H3K18cr mark and HDACs? Does it affect the expression level of cancer-associated genes? The author can perform ChIP-seq experiments using anti-H3K18cr and anti-HDACs antibodies to address those issues.

Our response: Again, the referee raises a series of very interesting question, whose answers, though, are beyond the scope of this current manuscript. We hope we can address these questions in the future.

Minor Comments:

1. Since the anti-H3K18ac and anti-H3K18cr antibodies bought from companies, there is no need to give unnecessary details of those antibodies. But commercial information of antibodies used in this work should be provided in “Material and Methods” section.

Our response: We followed this advice.

2. In page 54, line 887, it should be lysine acetylation instead of monomethylation.

Our response: This has been corrected.

3. In page 62, line 1064, it should be H3K4me3 instead of H34me3.

Our response: This has been corrected.

4. In the “Material and Methods” section, mass data analysis of histone crotonylation was missing.

Our response: We provide more detail and write (page 20, 476-485): “Mass tolerance for searches was set to a maximum of 6 parts per million (ppm) for peptide masses and 20 ppm for high-energy collisional dissociation (HCD) fragment ion masses. A maximum of three missed cleavages was allowed. In the search, we focused on lysine methylation and acylation, including as variable modifications: D3-acetylation (+45.0294 Da), D3-acetylation (+45.0294 Da) plus monomethylation (+14.016 Da), dimethylation (+28.031 Da), trimethylation (+42.046 Da), acetylation (+42.010 Da) and crotonylation (+68.074 Da). The use of high-accuracy criteria for HCD fragment ions tolerance (20 ppm) guarantee the capability to discriminate among other possible forms of acylation (e.g. lysine butyrylation (70.0418 Da) and β -hydroxybutyrylation (86.0367 Da)) therefore they were not included in the search.”

5. In page 54, “MS data analysis and relative abundance profiling” section, the explanation on D3-acetylation was missing.

Our response: We provide more detail and write (page 18, 437-442): “Chemical acetylation occurs on unmodified and monomethylated lysines and prevents trypsin digestion at these residues, thus producing a pattern of digestion similar to that obtained with the Arg-C protease (the so called “Arg-C like” in-gel digestion pattern). The resulting histone peptides display an optimal length for MS detection and enhanced hydrophobicity that increases their separation at ultra- pressure chromatographic regimes.”

Reviewer #2 (Remarks to the Author):

The authors have addressed my concerns and I thus recommend publication of the manuscript.

Reviewer #3 (Remarks to the Author):

This reviewer is satisfied with the author's responses to my critiques. The Wei et al paper (ref 28) does deal a blow to the novelty of some of the findings. However, even considering that paper this manuscript still presents several novel concepts. Most notably, the role of the microbiome in regulating Kcr, the differential effect of HDACi on Kcr vs Kac and the differential enzyme kinetics of HDAC towards Kcr vs Kac substrates.

Reviewer #4 (Remarks to the Author):

Fellow et al.

Histone decrotonylation in the intestinal epithelium by class I histone deacetylases

Reversible histone crotonylation is a chromatin modification linked to the cellular metabolism and has potential positive impact on transcription. Its biological and physiological role is largely unexplored. In an attempt to study tissue-specific effects of histone crotonylation the authors identified intestine and brain as tissues with particularly high levels of histone crotonylation. They show by ChIP-seq co-occupancy of regulatory regions by H3K18cr and H3K168ac. Based on HDAC inhibitor sensitivities and tests with purified enzymes they identify class I deacetylases HDAC1, HDAC2 and HDAC3 as relevant decrotonylases. Interestingly, short fatty acids such as butyrate, which are known natural HDAC inhibitors induce histone crotonylation. Based on the fact that butyrate and other short fatty acids are produced by microbes in the gut and intestine the authors show that depletion/reduction of the gut microbiome results in decreased histone crotonylation indicating a regulatory function of microbiome-produced short fatty acids on histone crotonylation and potentially on gene expression in intestinal cells.

Comments:

Given that this manuscript was previously reviewed by three experts I have only a major comment and few minor points. The mouse experiments (depletion of the gut microbiome) significantly add to the importance of the manuscript because they suggest a biological significance of histone crotonylation with respect to intestinal homeostasis and maybe also to cancer.

Minor points:

Figure 3: In addition to the inhibitory function on class I HDACs short fatty acids produced by gut microbes also regulate the protein level of HDACs. HDAC2 levels are lower in the presence of an intact microbiome and elevated upon treatment with antibiotics. Kraemer et al. (2003) have shown that the carboxylic acids valproic acid and butyrate reduce the stability of the HDAC2 enzyme. This should be mentioned.

Our response: We thank this reviewer for pointing out the Kraemer et al. paper, which is indeed very relevant. We write now in discussion (page 13, line 315-319): “In this context, it is interesting to note that a previous study has shown that the stability of HDAC2 is selectively reduced by the HDAC inhibitor valproic acid (a branched short chain fatty acid) or butyrate and this is mediated by Ubc8-RLIM targeted proteasomal degradation”.

Figure 6: The cell cycle regulation of histone crotonylation is interesting. Class I HDACs have been implicated in the deacetylation of de novo synthesized HAT B-acetylated core histones (for instance: Bashkara et al. 2010). The MS-275 experiments indicate that reduced histone crotonylation is not only due to dilution of existing histones by non-crotonylated de novo synthesized histones but by an active process mediated by class I HDACs. This is an interesting finding which could be discussed in the manuscript.

Our response: We thank this reviewer for these suggestions and provide now a deeper discussion of the finding of cell cycle regulation of histone crotonylation, taking the advice from this reviewer and also reviewer number 1. Please, see our response to major comment 3 of reviewer number 1.